# CERTIFIED DEFENSE ON THE FAIRNESS OF GRAPH NEURAL NETWORKS

## ABSTRACT

Graph Neural Networks (GNNs) have emerged as a prominent graph learning model in various graph-based tasks over the years. Nevertheless, due to the vulnerabilities of GNNs, it has been empirically proved that malicious attackers could easily corrupt the fairness level of their predictions by adding perturbations to the input graph data. In this paper, we take crucial steps to study a novel problem of certifiable defense on the fairness level of GNNs. Specifically, we propose a principled framework named ELEGANT and present a detailed theoretical certification analysis for the fairness of GNNs. ELEGANT takes *any* GNNs as its backbone, and the fairness level of such a backbone is theoretically impossible to be corrupted under certain perturbation budgets for attackers. Notably, ELEGANT does not have any assumption over the GNN structure or parameters, and does not require re-training the GNNs to realize certification. Hence it can serve as a plug-and-play framework for any optimized GNNs ready to be deployed. We verify the satisfactory effectiveness of ELEGANT in practice through extensive experiments on real-world datasets across different backbones of GNNs, where ELEGANT is also demonstrated to be beneficial for GNN debiasing.

## 1 INTRODUCTION

Graph Neural Networks (GNNs) have emerged among the most popular models to handle learning tasks on graphs (Kipf & Welling, 2017; Veličković et al., 2018) and made remarkable achievements in various domains (Feng et al., 2022; Li et al., 2022). Nevertheless, as GNNs are increasingly deployed in real-world decision-making scenarios, there has been an increasing societal concern on the fairness of GNN predictions. A primary reason is that most traditional GNNs do not consider fairness, and thus could exhibit bias against certain demographic subgroups. Here the demographic subgroups are usually divided by certain sensitive attributes, such as gender and race. To prevent GNNs from biased predictions, multiple recent studies have proposed fairness-aware GNNs (Agarwal et al., 2021; Dai & Wang, 2021; Kang et al., 2022a) such that potential bias could be mitigated.

Unfortunately, despite existing efforts towards fair GNNs, it remains difficult to prevent the corruption of their fairness level due to their common vulnerability of lacking adversarial robustness. In fact, malicious attackers can easily corrupt the fairness level of GNNs by perturbing the node attributes (i.e., changing the values of node attributes) and/or the graph structure (i.e., adding and deleting edges) (Hussain et al., 2022), which could lead to serious consequences in the test phase (Dai & Wang, 2021; Hussain et al., 2022). For example, GNNs have been leveraged to perform bail decision-making on the graph of defendants, where an edge between two defendants represents high profile similarity (Agarwal et al., 2021). Yet, by simply injecting adversarial links in the graph data, attackers can make GNNs deliver advantaged predictions for a subgroup (e.g., individuals with a certain nationality) while damaging the interest of others (Hussain et al., 2022). Hence achieving defense over the fairness of GNNs is crucial for the purpose of safe deployment.

It is worth noting that despite the abundant empirical defense strategies for GNNs (Zhang & Zitnik, 2020; Entezari et al., 2020; Jin & Zhang, 2019; Jin et al., 2020c; Wu et al., 2019b), they are always subsequently defeated by novel attacking techniques (Schuchardt et al., 2020; Carlini & Wagner, 2017), and the defense over the fairness of GNNs also faces the same problem. Therefore, an ideal way is to achieve certifiable defense on fairness (i.e., certified fairness defense). A few recent

works aim to certify the fairness for traditional deep learning models (Khedr & Shoukry, 2022; Kang et al., 2022b; Jin et al., 2022; Mangold et al., 2022; Borca-Tasciuc et al., 2022; Ruoss et al., 2020). Nevertheless, most of them require specially designed training strategies (Khedr & Shoukry, 2022; Jin et al., 2022; Ruoss et al., 2020) and thus cannot be directly applied to optimized GNNs ready to be deployed. More importantly, they mostly rely on assumptions on the optimization results (Khedr & Shoukry, 2022; Jin et al., 2022; Borca-Tasciuc et al., 2022; Ruoss et al., 2020) or data distributions (Kang et al., 2022b; Mangold et al., 2022) over a continuous input space. Hence they can hardly be generalized to GNNs due to the binary nature of the input graph topology. Several other works propose certifiable GNN defense approaches to achieve theoretical guarantee (Wang et al., 2021; Bojchevski & Günnemann, 2019; Bojchevski et al., 2020; Jin et al., 2020a; Zügner & Günnemann, 2019; 2020). However, they mainly focus on securing the GNN prediction for a certain individual node to ensure model utility, ignoring the fairness defense over the entire population. Therefore, despite the significance, the study in this field still remains in its infancy.

It is worth noting that achieving certifiable defense on the fairness of GNNs is a daunting task due to the following key challenges: (1) **Generality:** different types of GNNs could be designed and optimized for different real-world applications (Zhou et al., 2020). Correspondingly, our first challenge is to design a plug-and-play framework that can achieve certified defense on fairness for any optimized GNN models that are ready to be deployed. (2) **Vulnerability:** a plethora of existing studies have empirically verified that most GNNs are sensitive to input data perturbations (Zhang & Zitnik, 2020; Zügner et al., 2020; Xu et al., 2019). In other words, small input perturbations may cause significant changes in the GNN output. Hence our second challenge is to properly mitigate the common vulnerabilities of GNNs without changing its structure or re-training. (3) **Multi-Modality:** the input data of GNNs naturally bears multiple modalities. For example, there are node attributes and graph topology in the widely studied attributed networks. In practice, both data modalities may be perturbed by malicious attackers. Therefore, our third challenge is to achieve certified defenses of fairness on both data modalities for GNNs.

As an early attempt to address the aforementioned challenges, in this paper, we propose a principled framework named ELEGANT (c**E**tifiab**LE G**NNs over the f**A**ir**N**ess of Predic**T**ions). Specifically, we focus on the widely studied task of node classification and formulate a novel research problem of *Certifying GNN Classifiers on Fairness*. To handle the first challenge, we propose to develop ELEGANT on top of an optimized GNN model without any assumptions over its structure or parameters. Hence ELEGANT is able to serve as a plug-and-play framework for any optimized GNN model ready to be deployed. To handle the second challenge, we propose to leverage randomized smoothing (Wang et al., 2021; Cohen et al., 2019) to defend against malicious attacks, where most GNNs can then be more robust over the prediction fairness level. To handle the third challenge, we propose two different strategies working in a concurrent manner, such that certified defense against the attacks on both the node attributes (i.e., add and subtract attribute values) and graph topology (i.e., flip the existence of edges) can be realized. Finally, we evaluate the effectiveness of ELEGANT on multiple real-world network datasets. In summary, our contributions are three-fold: (1) **Problem Formulation.** We formulate and make an initial investigation on a novel research problem of *Certifying GNN Classifiers on Fairness*. (2) **Algorithm Design.** We propose a framework ELEGANT to achieve certified fairness defense against attacks on both node attributes and graph structure without relying on assumptions about any specific GNNs. (3) **Experimental Evaluation.** We perform comprehensive experiments on real-world datasets to verify the effectiveness of ELEGANT.

## 2 PROBLEM DEFINITION

**Preliminaries.** Let $\mathcal{G} = \{\mathcal{V}, \mathcal{E}\}$ be an undirected attributed network, where $\mathcal{V} = \{v_1, ..., v_n\}$ is the set of $n$ nodes; $\mathcal{E} \subseteq \mathcal{V} \times \mathcal{V}$ is the set of edges. Let $\boldsymbol{A} \in \{0, 1\}^{n \times n}$ and $\boldsymbol{X} \in \mathbb{R}^{n \times d}$ be the adjacency matrix and attribute matrix of $\mathcal{G}$, respectively. Assume each node in $\mathcal{G}$ represents an individual, and sensitive attribute $s$ divides the population into different demographic subgroups. We follow a widely studied setting (Agarwal et al., 2021; Dai & Wang, 2021) to assume the sensitive attribute is binary, i.e., $s \in \{0, 1\}$. We use $s_i$ to denote the value of the sensitive attribute for node $v_i$. In node classification tasks, we use $\mathcal{V}_{\text{trn}}$ and $\mathcal{V}_{\text{tst}}$ ($\mathcal{V}_{\text{trn}}, \mathcal{V}_{\text{tst}} \in \mathcal{V}$) to represent the training and test node set, respectively. We denote the GNN node classifier as $f_{\boldsymbol{\theta}}$ parameterized by $\boldsymbol{\theta}$. $f_{\boldsymbol{\theta}}$ takes $\boldsymbol{A}$ and $\boldsymbol{X}$ as

input, and outputs $\hat{Y}$ as the predictions for the nodes in $\mathcal{G}$. Each row in $\hat{Y}$ is a one-hot vector flagging the predicted class. We use $f_{\boldsymbol{\theta}^*}$ to denote the GNN with optimal parameter $\boldsymbol{\theta}^*$.

**Threat Model.** We focus on the attacking scenario of model evasion, i.e., the attack happens in the test phase. In particular, we assume that the victim model under attack is an optimized GNN node classifier $f_{\boldsymbol{\theta}^*}$. We follow a widely adopted setting (Bojchevski & Günnemann, 2019; Zügner & Günnemann, 2019; Ma et al., 2020; Mu et al., 2021) to assume that a subset of nodes $\mathcal{V}_{\text{vul}} \in \mathcal{V}_{\text{tst}}$ are vulnerable to attacks. Specifically, attackers may perturb their links (i.e., flip the edge existence) to other nodes and/or their node attributes (i.e., change their attribute values). We denote the perturbations on adjacency matrix as $\boldsymbol{A} \oplus \boldsymbol{\Delta_A}$. Here $\oplus$ denotes the element-wise XOR operator; $\boldsymbol{\Delta_A} \in \{0, 1\}^{n \times n}$ is the matrix representing the perturbations made by the attacker, where 1 only appears in rows and columns associated with the vulnerable nodes while 0 appears elsewhere. Correspondingly, in $\boldsymbol{\Delta_A}$, 1 entries represent edges that attackers intend to flip, while 0 entries are associated with edges that are not attacked. Similarly, we denote the perturbations on node attribute matrix as $\boldsymbol{X} + \boldsymbol{\Delta_X}$, where $\boldsymbol{\Delta_X} \in \mathbb{R}^{n \times n}$ is the matrix representing the perturbations made by the attacker. Usually, if the total magnitude of perturbations is within certain budgets (i.e., $\|\boldsymbol{\Delta_A}\|_0 \leq \epsilon_{\boldsymbol{A}}$ for $\boldsymbol{A}$ and $\|\boldsymbol{\Delta_X}\|_2 \leq \epsilon_{\boldsymbol{X}}$ for $\boldsymbol{X}$), the perturbations are regarded as unnoticeable. The goal of an attacker is to add unnoticeable perturbations to nodes in $\mathcal{V}_{\text{vul}}$, such that the GNN predictions for nodes in $\mathcal{V}_{\text{tst}}$ based on the perturbed graph exhibit as much bias as possible. In addition, we assume that the attacker has access to any information about the victim GNN (i.e., a white-box setting). This is the worst case in practice, which makes it even more challenging to achieve defense.

To defend against the aforementioned attacks, we aim to establish a node classifier on top of an optimized GNN backbone, such that this classifier, theoretically, will not exhibit more bias than a given threshold no matter what unnoticeable perturbations (i.e., perturbations within budgets) are added. We formally formulate the problem of *Certifying GNN Classifiers on Fairness* below.

**Problem 1.** *Certifying GNN Classifiers on Fairness. Given an attributed network $\mathcal{G}$, a test node set $\mathcal{V}_{\text{tst}}$, a vulnerable node set $\mathcal{V}_{\text{vul}} \in \mathcal{V}_{\text{tst}}$, a threshold $\eta$ for the exhibited bias, and an optimized GNN classifier $f_{\boldsymbol{\theta}^*}$, our goal is to achieve a classifier on top of $f_{\boldsymbol{\theta}^*}$ associated with budgets $\epsilon_{\boldsymbol{A}}$ and $\epsilon_{\boldsymbol{X}}$, such that this classifier will bear comparable utility with $f_{\boldsymbol{\theta}^*}$ but provably not exhibit more bias than $\eta$ on the nodes in $\mathcal{V}_{\text{tst}}$, no matter what unnoticeable node attributes and/or graph structure perturbations (i.e., perturbations within budgets) are made over the nodes in $\mathcal{V}_{\text{vul}}$.*

## 3 METHODOLOGY

Here we first introduce the modeling of attack and defense on the fairness of GNNs, then discuss how we achieve certified defense on node attributes. After that, we propose a strategy to achieve both types of certified defense (i.e., defense on node attributes and graph structure) at the same time. Finally, we introduce strategies to achieve the designed certified fairness defense for GNNs in practice.

### 3.1 BIAS INDICATOR FUNCTION

We first construct an indicator $g$ to mathematically model the attack and defense on the fairness of GNNs. Our rationale is to use $g$ to indicate whether the predictions of $f_{\boldsymbol{\theta}^*}$ exhibit a level of bias exceeding a given threshold. We present the formal definition below.

**Definition 1.** *(Bias Indicator Function) Given adjacency matrix $\boldsymbol{A}$ and node attribute matrix $\boldsymbol{X}$, a test node set $\mathcal{V}_{\text{tst}}$, a threshold $\eta$ for the exhibited bias, and an optimized GNN model $f_{\boldsymbol{\theta}^*}$, the bias indicator function is defined as $g(f_{\boldsymbol{\theta}^*}, \boldsymbol{A}, \boldsymbol{X}, \eta, \mathcal{V}_{\text{tst}}) = \mathbb{1}\left(\pi(f_{\boldsymbol{\theta}^*}(\boldsymbol{A}, \boldsymbol{X}), \mathcal{V}_{\text{tst}}) < \eta\right)$, where $\mathbb{1}(\cdot)$ takes an event as input and outputs 1 if the event happens (otherwise 0); $\pi(\cdot, \cdot)$ denotes any bias metric for GNN predictions (taken as its first parameter) over a set of nodes (taken as its second parameter). Traditional bias metrics include $\Delta_{SP}$ (Dai & Wang, 2021; Dwork et al., 2012) and $\Delta_{EO}$ (Dai & Wang, 2021; Hardt et al., 2016).*

Correspondingly, the goal of the attacker is to ensure that the indicator $g$ outputs 0 for an $\eta$ as large as possible, while the goal of certified defense is to ensure for a given threshold $\eta$, the indicator $g$ provably yields 1 as long as the attacks are within certain budgets. Note that a reasonable $\eta$ should

ensure that $g$ outputs 1 based on the clean graph data (i.e., graph data without any attacks). Below we first discuss the certified fairness defense over node attributes to maintain the output of $g$ as 1.

### 3.2 Certified Fairness Defense over Node Attributes

We now introduce how we achieve certified defense over the node attributes for the fairness of the predictions yielded by $f_{\boldsymbol{\theta}^*}$. Specifically, we propose to construct a smoothed bias indicator function $\tilde{g}_{\boldsymbol{X}}(f_{\boldsymbol{\theta}^*}, \boldsymbol{A}, \boldsymbol{X}, \mathcal{V}_{\text{vul}}, \eta)$ via adding Gaussian noise over the node attributes of vulnerable nodes in $\mathcal{V}_{\text{vul}}$. For simplicity, we use $\tilde{g}_{\boldsymbol{X}}(\boldsymbol{A}, \boldsymbol{X})$ to represent the smoothed bias indicator function over node attributes by omitting $\mathcal{V}_{\text{vul}}$, $f_{\boldsymbol{\theta}^*}$ and $\eta$. We formally define $\tilde{g}_{\boldsymbol{X}}$ below.

**Definition 2.** *(Bias Indicator with Node Attribute Smoothing) We define the bias indicator with smoothed node attributes over the nodes in $\mathcal{V}_{vul}$ as $\tilde{g}_{\boldsymbol{X}}(\boldsymbol{A}, \boldsymbol{X}) = \text{argmax}_{c \in \{0,1\}} \Pr(g(f_{\boldsymbol{\theta}^*}, \boldsymbol{A}, \boldsymbol{X} + \gamma_{\boldsymbol{X}}(\boldsymbol{\omega}_{\boldsymbol{X}}, \mathcal{V}_{vul}), \eta, \mathcal{V}_{tst}) = c)$. Here $\boldsymbol{\omega}_{\boldsymbol{X}}$ is a $(d \cdot |\mathcal{V}_{vul}|)$-dimensional vector, where each entry is a random variable following a Gaussian Distribution $\mathcal{N}(0, \sigma^2)$; $\gamma_{\boldsymbol{X}}(\cdot, \cdot)$ maps a vector (its first parameter) to an $(n \times d)$-dimensional matrix, where the vector values are assigned to rows with the indices indicated by a set of nodes (its second parameter) while other entries are zeros.*

We denote $\boldsymbol{\Gamma}_{\boldsymbol{X}} = \gamma_{\boldsymbol{X}}(\boldsymbol{\omega}_{\boldsymbol{X}}, \mathcal{V}_{\text{vul}})$ and $g(\boldsymbol{A}, \boldsymbol{X} + \boldsymbol{\Gamma}_{\boldsymbol{X}}) = g(f_{\boldsymbol{\theta}^*}, \boldsymbol{A}, \boldsymbol{X} + \gamma_{\boldsymbol{X}}(\boldsymbol{\omega}_{\boldsymbol{X}}, \mathcal{V}_{\text{vul}}), \eta, \mathcal{V}_{\text{tst}})$ below for simplicity. We are then able to derive the theoretical certification for the defense on fairness with the defined $\tilde{g}_{\boldsymbol{X}}$ in Definition 2. We now present the defense certification on fairness below.

**Theorem 1.** *(Certified Fairness Defense for Node Attributes) Denote the probability for $g(\boldsymbol{A}, \boldsymbol{X} + \boldsymbol{\Gamma}_{\boldsymbol{X}})$ to return class $c$ $(c \in \{0, 1\})$ as $P(c)$. Then $\tilde{g}_{\boldsymbol{X}}(\boldsymbol{A}, \boldsymbol{X})$ will provably return $\text{argmax}_{c \in \{0,1\}} P(c)$ for any perturbations (over the attributes of vulnerable nodes) within an $l_2$ radius $\tilde{\epsilon}_{\boldsymbol{X}} = \frac{\sigma}{2}\left(\Phi^{-1}(\max_{c \in \{0,1\}} P(c)) - \Phi^{-1}(\min_{c \in \{0,1\}} P(c))\right)$, where $\Phi^{-1}(\cdot)$ is the inverse of the standard Gaussian cumulative distribution function.*

Correspondingly, for an $\eta$ that enables $\max_{c \in \{0,1\}} P(c) = 1$, it is then safe to say that no matter what perturbations $\boldsymbol{\Delta}_{\boldsymbol{X}}$ are made on vulnerable nodes, as long as $\|\boldsymbol{\Delta}_{\boldsymbol{X}}\|_2 \leq \tilde{\epsilon}_{\boldsymbol{X}}$, the constructed $\tilde{g}_{\boldsymbol{X}}$ will provably not yield predictions for $\mathcal{V}_{\text{tst}}$ with a level of bias exceeding $\eta$. Nevertheless, it is worth noting that, in GNNs, perturbations may also be made on the structure of the vulnerable nodes, i.e., adding and/or deleting edges between these vulnerable nodes and any nodes in the graph. Hence it is also necessary to achieve certified defense against such structural attacks. Here we propose to also smooth the constructed $\tilde{g}_{\boldsymbol{X}}$ over the graph structure (of the vulnerable nodes) for the purpose of certified fairness defense on the graph structure. However, the adjacency matrix describing the graph structure is naturally binary, and thus should be smoothed in a different way.

### 3.3 Certified Fairness Defense over Node Attributes and Graph Structure

We then introduce achieving certified fairness defense against attacks on both node attributes and graph structure. We propose a strategy to leverage noise following Bernoulli distribution to smooth $\tilde{g}_{\boldsymbol{X}}$ over the rows and columns (due to symmetricity) associated with the vulnerable nodes in $\boldsymbol{A}$. In this way, we can smooth both the node attributes and graph structure for $g$ in a randomized manner, and we denote the constructed function as $\tilde{g}_{\boldsymbol{A},\boldsymbol{X}}$. We present the formal definition below.

**Definition 3.** *(Bias Indicator with Attribute-Structure Smoothing) We define the bias indicator function with smoothed node attributes and graph structure over the nodes in $\mathcal{V}_{vul}$ as $\tilde{g}_{\boldsymbol{A},\boldsymbol{X}}(\boldsymbol{A}, \boldsymbol{X}) = \text{argmax}_{c \in \{0,1\}} \Pr(\tilde{g}_{\boldsymbol{X}}(\boldsymbol{A} \oplus \gamma_{\boldsymbol{A}}(\boldsymbol{\omega}_{\boldsymbol{A}}, \mathcal{V}_{vul}), \boldsymbol{X}) = c)$. Here $\boldsymbol{\omega}_{\boldsymbol{A}}$ is an $(n \cdot |\mathcal{V}_{vul}|)$-dimensional random variable, where each dimension takes 0 and 1 with the probability of $\beta$ ($0.5 < \beta \leq 1$) and $1 - \beta$, respectively; function $\gamma_{\boldsymbol{A}}(\cdot, \cdot)$ maps a vector (its first parameter) to a symmetric $(n \times n)$-dimensional matrix, where the vector values are assigned to rows whose indices associated with the indices of a set of nodes (its second parameter) and then mirrored to the corresponding columns, while other values are left as zeros.*

We let $\boldsymbol{\Gamma}_{\boldsymbol{A}} = \gamma_{\boldsymbol{A}}(\boldsymbol{\omega}_{\boldsymbol{A}}, \mathcal{V}_{\text{vul}})$ below for simplicity. To better illustrate how classifier $\tilde{g}_{\boldsymbol{A},\boldsymbol{X}}$ achieves certified fairness defense over both data modalities of an attributed network, we provide an exemplary case in Figure 1. Here we assume node $v_i \in \mathcal{V}_{\text{vul}}$. Considering the high dimensionality of node attributes and adjacency matrix, we only analyze two entries $\boldsymbol{X}_{i,j}$ and $\boldsymbol{A}_{i,j}$

and omit other entries after noise for simplicity. Here the superscript $(i,j)$ represents the $i$-th row and $j$-th column of a matrix. Under binary noise, entry $\boldsymbol{A}_{i,j}$ only has two possible values, i.e., $\boldsymbol{A}_{i,j} \oplus 0$ and $\boldsymbol{A}_{i,j} \oplus 1$. We denote the two cases as Case (1) and Case (2), respectively. We assume that the area where $g$ returns 1 in the span of the two input random entries of $g$ (i.e., $\boldsymbol{X}_{i,j}$ and $\boldsymbol{A}_{i,j}$ under random noise) is an ellipse (marked out with green), where the decision boundary is marked out with deep green. In Case (1), $\boldsymbol{X}_{i,j}$ under random noise follows a Gaussian distribution, whose probability density function is marked out as deep red.

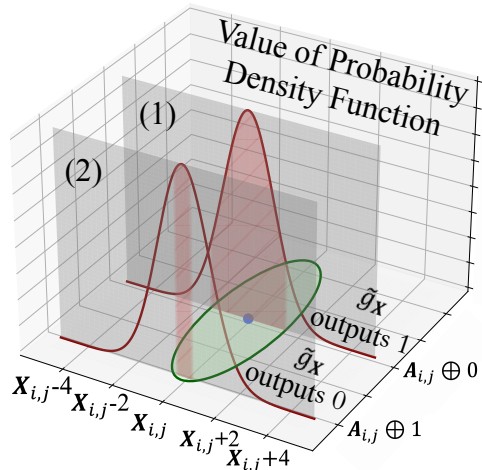

Figure 1: An example illustrating how ELE-GANT works in the input space.

We assume that, in this case, the integral of the probability density function within the range of the ellipse (marked out with shallow red) is larger than 0.5. Correspondingly, according to Definition 2, $\tilde{g}_{\boldsymbol{X}}$ returns 1 in this case. In Case (2), we similarly mark out the probability density function and the area used for integral within the range of the ellipse. We assume that in this case, the integral is smaller than 0.5, and thus $\tilde{g}_{\boldsymbol{X}}$ returns 0. Note that to compute the output of $\tilde{g}_{\boldsymbol{A},\boldsymbol{X}}$, we need to identify the output of $\tilde{g}_{\boldsymbol{X}}$ with the largest probability. Notice that $\beta > 0.5$, we have that $\boldsymbol{A}_{i,j} \oplus 0$ happens with a larger probability than $\boldsymbol{A}_{i,j} \oplus 1$. Therefore, $\tilde{g}_{\boldsymbol{A},\boldsymbol{X}}$ outputs 1 in this example. In other words, the bias level of the predictions of $f_{\boldsymbol{\theta}^*}$ is satisfying (i.e., smaller than $\eta$) based on $\tilde{g}_{\boldsymbol{A},\boldsymbol{X}}$.

Below we introduce a desirable property of $\tilde{g}_{\boldsymbol{A},\boldsymbol{X}}$, i.e., certified fairness defense associated with tractable budgets over both node attributes and graph topology can be achieved.

**Lemma 1.** *(Perturbation-Invariant Budgets Existence) There exist tractable budgets $\epsilon_{\boldsymbol{A}}$ and $\epsilon_{\boldsymbol{X}}$, such that for any perturbations made over the node attributes and graph structure of the vulnerable nodes within $\epsilon_{\boldsymbol{A}}$ and $\epsilon_{\boldsymbol{X}}$, $\tilde{g}_{\boldsymbol{A},\boldsymbol{X}}$ provably maintains the same classification results.*

Correspondingly, for an $\eta$ that enables $\tilde{g}_{\boldsymbol{A},\boldsymbol{X}}$ to return 1, we are then able to achieve certified fairness defense over $\tilde{g}_{\boldsymbol{A},\boldsymbol{X}}$ against perturbations on both node attributes and graph structure. Below we derive the certified fairness defense budgets over the graph structure $\epsilon_{\boldsymbol{A}}$ and node attributes $\epsilon_{\boldsymbol{X}}$ for $\tilde{g}_{\boldsymbol{A},\boldsymbol{X}}$. We first introduce the derivation of $\epsilon_{\boldsymbol{A}}$. Here, our rationale is: considering that $\tilde{g}_{\boldsymbol{A},\boldsymbol{X}}$ is a binary classifier, we need to ensure that under structure attacks, the probability of $\tilde{g}_{\boldsymbol{X}}$ returning 1 (denoted as $\Pr(\tilde{g}_{\boldsymbol{X}}(\boldsymbol{A} \oplus \boldsymbol{\Delta}_{\boldsymbol{A}} \oplus \boldsymbol{\Gamma}_{\boldsymbol{A}}, \boldsymbol{X}) = 1)$) is provably greater than 0.5, such that $\tilde{g}_{\boldsymbol{A},\boldsymbol{X}}$ will still return 1. To this end, we propose to derive a lower bound of $\Pr(\tilde{g}_{\boldsymbol{X}}(\boldsymbol{A} \oplus \boldsymbol{\Delta}_{\boldsymbol{A}} \oplus \boldsymbol{\Gamma}_{\boldsymbol{A}}, \boldsymbol{X}) = 1)$, which we denote as $\underline{P_{\tilde{g}_{\boldsymbol{X}}=1}}$. Finally, we identify the largest perturbation size that keeps such a lower bound larger than 0.5, and the identified perturbation size is then the graph structure perturbation budget. We present the lower bound of $\Pr(\tilde{g}_{\boldsymbol{X}}(\boldsymbol{A} \oplus \boldsymbol{\Delta}_{\boldsymbol{A}} \oplus \boldsymbol{\Gamma}_{\boldsymbol{A}}, \boldsymbol{X}) = 1)$ below.

**Lemma 2.** *(Positive Probability Bound Under Noises) There exists a tractable $\underline{P_{\tilde{g}_{\boldsymbol{X}}=1}} \in (0,1)$, such that $\Pr(\tilde{g}_{\boldsymbol{X}}(\boldsymbol{A} \oplus \boldsymbol{\Delta}_{\boldsymbol{A}} \oplus \boldsymbol{\Gamma}_{\boldsymbol{A}}, \boldsymbol{X}) = 1) \geq \underline{P_{\tilde{g}_{\boldsymbol{X}}=1}}$.*

To derive the perturbation budget $\epsilon_{\boldsymbol{A}}$, we only need to find a $\boldsymbol{\Delta}_{\boldsymbol{A}}$ with the largest $l_0$-norm that still enables $\underline{P_{\tilde{g}_{\boldsymbol{X}}=1}}$ to be greater than 0.5 (according to Definition 3). Correspondingly, we derive the theoretical perturbation-invariant budget $\epsilon_{\boldsymbol{A}}$ in Theorem 2 below.

**Theorem 2.** *(Certified Defense Budget for Structure Perturbations) The certified defense budget over the graph structure $\epsilon_{\boldsymbol{A}}$ for $\tilde{g}_{\boldsymbol{A},\boldsymbol{X}}$ is given as*
$$\epsilon_{\boldsymbol{A}} = \max \tilde{\epsilon}_{\boldsymbol{A}}, \ s.t. \ \underline{P_{\tilde{g}_{\boldsymbol{X}}=1}} > 0.5, \ \forall \ \|\boldsymbol{\Delta}_{\boldsymbol{A}}\|_0 \leq \tilde{\epsilon}_{\boldsymbol{A}}. \tag{1}$$

To solve the optimization problem in Equation (1), we introduce Theorem 3 to compute $\underline{P_{\tilde{g}_{\boldsymbol{X}}=1}}$.

**Theorem 3.** *(Positive Probability Lower Bound) We have* $\underline{P_{\tilde{g}_X=1}} = \Pr(\boldsymbol{A} \oplus \boldsymbol{\Delta}_{\boldsymbol{A}} \oplus \boldsymbol{\Gamma}_{\boldsymbol{A}} \in \mathcal{H})$. *Here* $\mathcal{H} = \cup_{i=\mu+1}^{n \cdot |\mathcal{V}_{vul}|} \mathcal{H}_i \cup \mathcal{H}'_{\mu}$; $\mathcal{H}_i$ *is given by*

$$\mathcal{H}_i = \left\{ \bar{\boldsymbol{A}} : \frac{\Pr(\boldsymbol{A} \oplus \boldsymbol{\Gamma}_{\boldsymbol{A}} = \bar{\boldsymbol{A}})}{\Pr(\boldsymbol{A} \oplus \boldsymbol{\Delta}_{\boldsymbol{A}} \oplus \boldsymbol{\Gamma}_{\boldsymbol{A}} = \bar{\boldsymbol{A}})} = \left( \frac{\beta}{1-\beta} \right)^i, \forall v_i \in \mathcal{V} \setminus \mathcal{V}_{vul}, \|\bar{\boldsymbol{A}}_i - \boldsymbol{A}_i\|_0 = 0 \right\};$$

*and* $\mu$ *is defined over the optimization problem of* $\operatorname{argmax}_{-n \cdot |\mathcal{V}_{vul}| \leq j \leq n \cdot |\mathcal{V}_{vul}|} j$, *s.t.* $\Pr(\tilde{g}_{\boldsymbol{X}}(\boldsymbol{A} \oplus \boldsymbol{\Gamma}_{\boldsymbol{A}}, \boldsymbol{X}) = 1) \leq \Pr\left( \boldsymbol{A} \oplus \boldsymbol{\Gamma}_{\boldsymbol{A}} \in \cup_{k=j}^{n \cdot |\mathcal{V}_{vul}|} \mathcal{H}_j \right)$. *Here* $\mathcal{H}'_{\mu}$ *is any subregion of* $\mathcal{H}_{\mu}$ *that satisfies* $\Pr(\boldsymbol{A} \oplus \boldsymbol{\Gamma}_{\boldsymbol{A}} \in \mathcal{H}'_{\mu}) = \Pr(\tilde{g}_{\boldsymbol{X}}(\boldsymbol{A} \oplus \boldsymbol{\Gamma}_{\boldsymbol{A}}, \boldsymbol{X}) = 1) - \Pr\left( \boldsymbol{A} \oplus \boldsymbol{\Gamma}_{\boldsymbol{A}} \in \cup_{k=j}^{n \cdot |\mathcal{V}_{vul}|} \mathcal{H}_j \right)$.

We provide detailed steps to solve the optimization problem given in Equation (1) in Appendix. Now we introduce the theoretical analysis of how to derive $\epsilon_{\boldsymbol{X}}$ in Theorem 4.

**Theorem 4.** *(Certified Defense Budget over Node Attributes) Denote* $\bar{\mathcal{A}}$ *as the set of all possible* $(n \times n)$-matrices, where entries in rows whose indices associate with those vulnerable nodes may take 1 or 0, while other entries are zeros. The certified defense budget $\epsilon_{\boldsymbol{X}}$ for $\tilde{g}_{\boldsymbol{A}, \boldsymbol{X}}$ is given as $\epsilon_{\boldsymbol{X}} = \min\{\tilde{\epsilon_{\boldsymbol{X}}} : \tilde{\epsilon_{\boldsymbol{X}}}$ is derived with classifier $\tilde{g}_{\boldsymbol{X}}(\boldsymbol{A} \oplus \boldsymbol{\Gamma}_{\boldsymbol{A}}, \boldsymbol{X})$, where $\boldsymbol{\Gamma}_{\boldsymbol{A}} \in \bar{\mathcal{A}}\}$.

### 3.4 CERTIFICATION IN PRACTICE

**Estimating the Predicted Label Probabilities.** According to Definition 3, it is necessary to obtain $\Pr(\tilde{g}_{\boldsymbol{X}}(\boldsymbol{A} \oplus \boldsymbol{\Gamma}_{\boldsymbol{A}}, \boldsymbol{X}) = c)$ $(c \in \{0, 1\})$ to determine the output of classifier $\tilde{g}_{\boldsymbol{X}}$. We propose to leverage a Monte Carlo method to estimate such a probability. Specifically, we first randomly pick $N$ samples of $\boldsymbol{\Gamma}_{\boldsymbol{A}}$ as $\bar{\mathcal{A}}'$ $(\bar{\mathcal{A}}' \subset \bar{\mathcal{A}})$. Considering the output of $\tilde{g}_{\boldsymbol{X}}$ is binary, we then follow a common strategy (Cohen et al., 2019) to consider this problem as a parameter estimation of a Binomial distribution: we first count the number of returned label 1 and 0 under noise as $N_1$ and $N_0$ $(N_1 + N_0 = N)$; then we choose a confidence level $1 - \alpha$ and take the $\alpha$-th quantile of the beta distribution with parameters $N_1$ and $N_0$ as the estimated probability lower bound for returning label $c = 1$. We proved that all theoretical analysis still holds true for such an estimation in Appendix. We follow a similar strategy to estimate the probability lower bound of yielding 1 for $g(\boldsymbol{A}, \boldsymbol{X} + \boldsymbol{\Gamma}_{\boldsymbol{X}})$.

**Obtaining Fair Classification Results.** After achieving certified fairness defense based on $\tilde{g}_{\boldsymbol{A}, \boldsymbol{X}}$, we also need to obtain the corresponding node classification results (given by $f_{\boldsymbol{\theta}^*}$) over $\mathcal{V}_{\text{tst}}$. We propose to collect all classification results associated with the sampled $\boldsymbol{\Gamma}'_{\boldsymbol{A}} \in \bar{\mathcal{A}}'$ that leads to an estimated lower bound of $\Pr(\tilde{g}_{\boldsymbol{X}}(\boldsymbol{A} \oplus \boldsymbol{\Gamma}'_{\boldsymbol{A}}, \boldsymbol{X}) = 1)$ to be larger than 0.5 as $\hat{\mathcal{Y}}'$. Here $\hat{\mathcal{Y}}'$ is a set of output matrices of $f_{\boldsymbol{\theta}^*}$, where each matrix consists of the one-hot output classification results (as each row in the matrix) for all nodes. We propose to take $\operatorname{argmin}_{\hat{\boldsymbol{Y}}'} \pi(\hat{\boldsymbol{Y}}', \mathcal{V}_{\text{tst}}), s.t. \hat{\boldsymbol{Y}}' \in \hat{\mathcal{Y}}'$ as the final node classification results. Correspondingly, consider $\Pr(\tilde{g}_{\boldsymbol{X}}(\boldsymbol{A} \oplus \boldsymbol{\Gamma}'_{\boldsymbol{A}}, \boldsymbol{X}) = 1)$ falls into the confidence interval characterized by $1 - \alpha$, we have a neat probabilistic theoretical guarantee below.

**Proposition 1.** *(Probabilistic Guarantee for the Fairness Level of Node Classification). For* $\hat{\boldsymbol{Y}} = \operatorname{argmin}_{\hat{\boldsymbol{Y}}'} \pi(\hat{\boldsymbol{Y}}', \mathcal{V}_{tst}), s.t. \hat{\boldsymbol{Y}}' \in \hat{\mathcal{Y}}'$, we have $\Pr(\pi(\hat{\boldsymbol{Y}}, \mathcal{V}_{tst}) > \eta) < 0.5^{|\hat{\mathcal{Y}}'|}$.

Note that for a large enough sample size $N$, the cardinality of $\hat{\mathcal{Y}}'$ also tends to be large in practice. Hence it is safe to argue that $\Pr(\pi(\hat{\boldsymbol{Y}}, \mathcal{V}_{\text{tst}}) > \eta)$ tends to be small enough. In other words, we have a probability that is large enough to obtain results with a bias level lower than threshold $\eta$.

**Calculation of Perturbation Budgets.** We calculate $\epsilon_{\boldsymbol{A}}$ by solving the optimization problem given in Equation (1), and we provide the completed procedure in Appendix. For $\epsilon_{\boldsymbol{X}}$, we utilize a Monte Carlo method to estimate its value. More specifically, we leverage $\min\{\tilde{\epsilon_{\boldsymbol{X}}} : \tilde{\epsilon_{\boldsymbol{X}}}$ is derived with classifier $\tilde{g}_{\boldsymbol{X}}(\boldsymbol{A} \oplus \boldsymbol{\Gamma}'_{\boldsymbol{A}}, \boldsymbol{X})$, where $\boldsymbol{\Gamma}'_{\boldsymbol{A}} \in \bar{\mathcal{A}}'\}$ to estimate the value of $\epsilon_{\boldsymbol{X}}$.

## 4 EXPERIMENTAL EVALUATIONS

In this section, we aim to answer three research questions: **RQ1**: How well does ELEGANT perform in achieving certified fairness defense? **RQ2**: How does ELEGANT perform under fairness attacks compared to other popular fairness-aware GNNs? **RQ3**: How does ELEGANT perform under

different settings of parameters? We present the main experimental settings and representative results in this section due to space limits. Detailed settings and supplementary experiments are in Appendix.

## 4.1 EXPERIMENTAL SETTINGS

**Downstream Task and Datasets.** We focus on the widely studied node classification task, which is one of the most representative tasks in the domain of learning on graphs. We adopt three real-world network datasets that are widely used to perform studies on the fairness of GNNs, namely German Credit (Agarwal et al., 2021; Asuncion & Newman, 2007), Recidivism (Agarwal et al., 2021; Jordan & Freiburger, 2015), and Credit Defaulter (Agarwal et al., 2021; Yeh & Lien, 2009). We provide their basic information, including how these datasets are built and their statistics, in Appendix.

**Evaluation Metrics.** We perform evaluation from three main perspectives, including model utility, fairness, and certified defense. To evaluate utility, we adopt the node classification accuracy. To evaluate fairness, we adopt the widely used metrics $\Delta_{\text{SP}}$ (measuring bias under *Statistical Parity*) and $\Delta_{\text{EO}}$ (measuring bias under *Equal Opportunity*). To evaluate certified defense, we extend a traditional metric named *Certified Accuracy* (Wang et al., 2021; Cohen et al., 2019) in our experiments, and we name it as *Fairness Certification Rate* (FCR). Specifically, existing GNN certification works mainly focus on a certain individual node, and utilize certified accuracy to measure the ratio of nodes that are correctly classified and also successfully certified out of all test nodes (Wang et al., 2021). In this paper, however, we perform certified (fairness) defense for individuals over an entire test set (instead of for any specific individual). Accordingly, we propose to sample multiple test sets out of nodes that are not involved in the training and validation set. Then we perform certified fairness defense for all sampled test sets, and utilize the ratio of test sets that are successfully certified over all sampled sets as the metric of certified defense. The rationale of FCR is leveraging a Monte Carlo method to estimate the probability of being successfully certified for a randomly sampled test node set.

**GNN Backbones and Baselines.** Note that ELEGANT serves as a plug-and-play framework for any optimized GNNs ready to be deployed. To evaluate the generality of ELEGANT across GNNs, we adopt three of the most representative GNNs spanning across simple and complex ones, namely Graph Sample and Aggregate Networks (Hamilton et al., 2017) (GraphSAGE), Graph Convolutional Networks (Kipf & Welling, 2017) (GCN), and Jumping Knowledge Networks (JK). Note that to the best of our knowledge, existing works on fairness certification cannot certify the attacks over two data modalities (i.e., continuous node attributes and binary graph topology) at the same time, and thus cannot be naively generalized onto GNNs. Hence we compare the usability of GNNs before and after certification with ELEGANT. Moreover, we also adopt two popular fairness-aware GNNs as baselines to evaluate bias mitigation, including FairGNN (Dai & Wang, 2021) and NIFTY (Agarwal et al., 2021). Specifically, FairGNN utilizes adversarial learning to debias node embeddings, while NIFTY designs regularizations to debias node embeddings.

**Threat Models.** We propose to evaluate the performance of ELEGANT and other fairness-aware GNN models under actual attacks on fairness. We first introduce the threat model over graph structure. To the best of our knowledge, FA-GNN (Hussain et al., 2022) is the only work that performs graph structure attacks targeting the fairness of GNNs. Hence we adopt FA-GNN to attack graph structure. In terms of node attributes, to the best of our knowledge, no existing work has made any explorations. Hence we directly utilize gradient ascend to perform attacks. Specifically, after structure attacks have been performed, we identify the top-ranked node attribute elements (out of the node attribute matrix) that positively influence the exhibited bias the most via gradient ascend. For any given budget (of attacks) on node attributes, we add perturbations to these elements in proportion to their gradients.

## 4.2 RQ1: FAIRNESS CERTIFICATION EFFECTIVENESS

To answer RQ1, we investigate the performance of different GNNs after certification across different real-world attributed network datasets over FCR, utility, and fairness. We present the experimental results across three GNN backbones and three real-world attributed network datasets in Table 1. Here bias is measured with $\Delta_{\text{SP}}$, and we have similar observations on $\Delta_{\text{EO}}$. We summarize the main observations as follows: (1) **Fairness Certification Rate (FCR).** We observe that ELEGANT realizes values of FCR around or even higher than 90% for all three GNN backbones and three attributed

Table 1: Comparison between vanilla GNNs and certified GNNs under ELEGANT over three popular GNNs across three real-world datasets. Here ACC denotes node classification accuracy, and E- prefix marks out the GNNs under ELEGANT with certification. ↑ denotes the larger, the better; ↓ represents the opposite. Numerical values are in percentage, and the best ones are in bold.

| | German Credit | | | Recidivism | | | Credit Defaulter | | |
|---|---|---|---|---|---|---|---|---|---|
| | ACC (↑) | Bias (↓) | FCR (↑) | ACC (↑) | Bias (↓) | FCR (↑) | ACC (↑) | Bias (↓) | FCR (↑) |
| SAGE | 67.3 $_{\pm2.14}$ | 50.6 $_{\pm15.9}$ | N/A | 89.8 $_{\pm0.66}$ | 9.36 $_{\pm3.15}$ | N/A | **75.9** $_{\pm2.18}$ | 13.0 $_{\pm4.01}$ | N/A |
| E-SAGE | **71.0** $_{\pm1.27}$ | **16.3** $_{\pm10.9}$ | 98.7 $_{\pm1.89}$ | **89.9** $_{\pm0.90}$ | **6.39** $_{\pm2.85}$ | 94.3 $_{\pm6.65}$ | 73.4 $_{\pm0.50}$ | **8.94** $_{\pm0.99}$ | 94.3 $_{\pm3.30}$ |
| GCN | **59.6** $_{\pm3.64}$ | 37.4 $_{\pm3.24}$ | N/A | **90.5** $_{\pm0.73}$ | 10.1 $_{\pm3.01}$ | N/A | **65.8** $_{\pm0.29}$ | 11.1 $_{\pm3.22}$ | N/A |
| E-GCN | 58.2 $_{\pm1.82}$ | **3.52** $_{\pm3.77}$ | 96.3 $_{\pm1.89}$ | 89.6 $_{\pm0.74}$ | **9.56** $_{\pm3.22}$ | 96.0 $_{\pm3.56}$ | 65.2 $_{\pm0.99}$ | **7.28** $_{\pm1.46}$ | 92.7 $_{\pm5.19}$ |
| JK | **63.3** $_{\pm4.11}$ | 41.2 $_{\pm18.1}$ | N/A | **91.9** $_{\pm0.54}$ | 10.1 $_{\pm3.15}$ | N/A | 76.6 $_{\pm0.69}$ | 9.24 $_{\pm0.60}$ | N/A |
| E-JK | 62.3 $_{\pm4.07}$ | **22.4** $_{\pm1.95}$ | 97.0 $_{\pm3.00}$ | 89.3 $_{\pm0.33}$ | **6.26** $_{\pm2.78}$ | 89.5 $_{\pm10.5}$ | **77.7** $_{\pm0.27}$ | **3.37** $_{\pm2.64}$ | 99.3 $_{\pm0.47}$ |

network datasets, especially for the German Credit dataset, where vanilla GNNs tend to exhibit a high level of bias. The corresponding intuition is that, for nodes in any randomly sampled test set, we have a probability around or higher than 90% to successfully certify the fairness level of the predictions yielded by the GNN model with our proposed framework ELEGANT. Hence ELEGANT achieves a satisfying fairness certification rate across all adopted GNN backbones and datasets. (2) **Utility.** We found that compared with those vanilla GNN backbones, certified GNNs with ELEGANT also exhibit comparable and even higher node classification accuracy values in all cases. Hence we conclude that our proposed framework ELEGANT does not significantly jeopardize the utility of the vanilla GNN models, and those certified GNNs with ELEGANT still bear a high level of usability in terms of node classification accuracy. (3) **Fairness.** Although the goal of ELEGANT is not debiasing GNNs, we observe that certified GNNs with ELEGANT achieve better performances in all cases in terms of algorithmic fairness compared with those vanilla GNNs. This demonstrates that the proposed framework ELEGANT also contributes to bias mitigation. We conjecture that such an advantage of debiasing could be a mixed result of (1) adding random noise on node attributes and graph topology (as in Section 3.2 and Section 3.3) and (2) the proposed strategy of obtaining fair classification results (as in Section 3.4). We provide a more detailed analysis in Appendix B.9.

### 4.3 RQ2: FAIRNESS CERTIFICATION UNDER ATTACKS

To answer RQ2, we perform attacks on the fairness of GCN, E-GCN, FairGNN (with a GCN backbone), and NIFTY (with a GCN backbone). Considering the large size of the quadratic space spanned by the sizes of perturbations $\Delta_A$ and $\Delta_X$, we present the evaluation under four representative $(\|\Delta_A\|_0, \|\Delta_X\|_2)$ pairs. We set the threshold for bias $\eta$ to be 50% higher than the fairness level of the vanilla GCN model on clean data, since it empirically helps to achieve a high certification success rate under large perturbations.

We present the fairness levels of the four models in terms of $\Delta_{\text{EO}}$ in Figure 2. Note that we utilize a vanilla GCN to predict the labels for test nodes to obtain the classification results discussed in Section 3.4, and we also have similar observations on other GNNs/datasets. (1) **Fairness.** We found that the GCN model with the proposed framework ELEGANT achieves the lowest level of bias in all cases of fairness attacks. This observation is consistent with the superiority in fairness found in Table 1, which demonstrates that the fairness superiority of ELEGANT maintains even under attacks within a wide range of attacking perturbation sizes. (2) **Certification on Fairness.** We now compare the performance of E-GCN across different attacking perturbation sizes. We observed that under relatively small attacking perturbation sizes, i.e., $(2^0, 10^{-1})$, $(2^1, 10^0)$, and $(2^2, 10^1)$, ELEGANT successfully achieves certification over fairness, and the bias level increases slowly as the size of attacks increases. Under relatively large attacking perturbation size, i.e., $(2^3, 10^2)$, although the attacking budgets go beyond the certified budgets, GCN under ELEGANT still exhibits a fairness

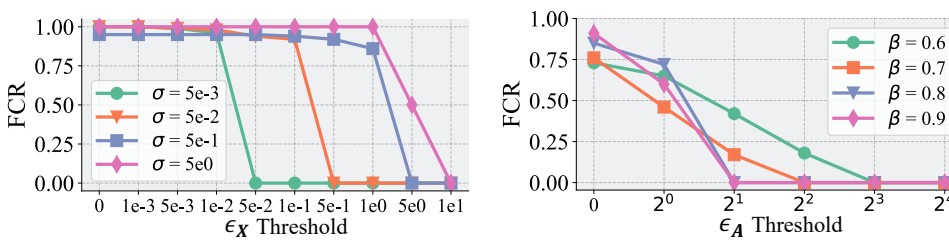

(a) FCR of certification for $\sigma$ over node attributes    (b) FCR of certification for $\beta$ over graph topology

Figure 3: Parameter study of $\sigma$ over $\epsilon_X$ (a) and $\beta$ over $\epsilon_A$ (b). Experimental results are presented based on GCN over German credit and Credit Defaulter for (a) and (b), respectively. Similar tendencies can also be observed based on other GNNs and datasets.

level far lower than the given bias threshold $\eta$, and the fairness superiority maintains. Hence the adopted estimation strategies are safe in achieving fairness certification.

## 4.4 RQ3: PARAMETER STUDY

To answer RQ3, we propose to perform parameter study focusing on two most critical parameters, $\sigma$ and $\beta$. To examine how $\sigma$ and $\beta$ influence the effectiveness of ELEGANT in terms of both FCR and certified defense budgets, we set numerical ranges for $\epsilon_X$ (from 0 to 1e1) and $\epsilon_A$ (from 0 to $2^4$) and divide the two ranges into grids. In both ranges, we consider the dividing values of the grids as thresholds for certification budgets. In other words, under each threshold, we only consider the test sets with the corresponding certified defense budget being larger than this threshold as successfully certified ones, and the values of FCR are re-computed accordingly. Our rationale here is that with the thresholds (for $\epsilon_X$ and $\epsilon_A$) increasing, if FCR reduces slowly, this demonstrates that most success-

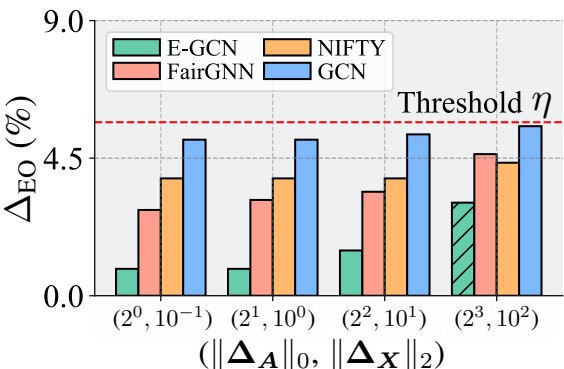

Figure 2: The bias levels of GCN, E-GCN, FairGNN, and NIFTY under fairness attacks on German Credit. The shaded bar indicates that certified budget $\epsilon_A \leq \|\Delta_A\|_0$ or $\epsilon_X \leq \|\Delta_X\|_2$. The y-axis is in logarithmic scale for better visualization purposes.

fully certified test sets are associated with large certified defense budgets. However, if FCR reduces fast, then most successfully certified test sets only bear small certified defense budgets.

Here we present the experimental results of $\sigma$ and $\beta$ with the most widely used GCN model based on German Credit in Figure 3(a) and Credit Defaulter in Figure 3(b), respectively. We also have similar observations on other GNNs and datasets. We summarize the main observations as follows: (1) **Analysis on $\sigma$.** We observe that most cases with larger $\sigma$ are associated with a larger FCR compared with the cases where $\sigma$ is relatively small. In other words, larger values of $\sigma$ typically make FCR reduce slower w.r.t. the increasing of $\epsilon_X$ threshold. This indicates that increasing the value of $\sigma$ helps realize larger certified defense budgets on node attributes, i.e., the increase of $\sigma$ dominates the tendency of $\epsilon_X$ given in Theorem 4. Nevertheless, it is worth mentioning that if $\sigma$ is too large, the information encoded in the node attributes could be swamped by the Gaussian noise and finally corrupt the classification accuracy. Hence moderately large values for $\sigma$, e.g., 5e-1 and 5e0, are recommended. (2) **Analysis on $\beta$.** We found that (1) for cases with relatively large $\beta$ (e.g., 0.8 and 0.9), the FCR also tends to be larger (compared with cases where $\beta$ is smaller) at $\epsilon_A$ threshold being 0. Such a tendency is reasonable, since in these cases, the expected magnitude of the added Bernoulli noise is small. Correspondingly, GNNs under ELEGANT perform similarly to vanilla GNNs, and thus an $\eta$ larger than the bias level of vanilla GNNs is easier to be satisfied (compared with cases under smaller values of $\beta$); (2) for cases with relatively large $\beta$, the value of FCR reduces faster (w.r.t. $\epsilon_A$ threshold) than cases where $\beta$ is smaller. Therefore, we recommend that for any test set of nodes:

(1) if the primary goal is to achieve certification with a high probability, then larger values for $\beta$ (e.g., 0.8 and 0.9) would be preferred; (2) if the goal is to achieve certification with larger certified defense budgets on the graph topology, then smaller values for $\beta$ (e.g., 0.6 and 0.7) should be selected.

## 5 RELATED WORK

**Algorithmic Fairness in GNNs.** Existing GNN works on fairness mainly focus on group fairness and individual fairness (Dong et al., 2022b). Specifically, group fairness requires that each demographic subgroup (divided by sensitive attributes such as gender and race) in the graph should have their fair share of interest based on predictions (M. et al., 2021). Adversarial training is among the most popular strategies (Dai & Wang, 2021; Dong et al., 2022b). In addition, regularization (Agarwal et al., 2021; Fan et al., 2021; Zhang et al., 2021), topology modification (Dong et al., 2022a; Spinelli et al., 2021), and orthogonal projection (Palowitch & Perozzi, 2020) are also commonly used strategies. On the other hand, individual fairness it requires that similar individuals should be treated similarly (Dwork et al., 2012), where such similarity may be determined in different ways (Kang et al., 2020; Dong et al., 2021). Designing optimization regularization terms to promote individual fairness for GNNs is a common strategy (Fan et al., 2021; Dong et al., 2021; Song et al., 2022). Nevertheless, despite the research advancements in the field of algorithmic fairness on GNNs, the adversarial defense against fairness attacks still remains in its infancy and has not been thoroughly explored. To the best of our knowledge, our paper serves as the first comprehensive study dedicated to addressing this important research problem, paving the way for future investigations in this under-explored area.

**GNN Defense Against Attacks.** Existing works on GNN defense are mainly categorized into four mainstreams, namely adversarial training (Xu et al., 2019; Dai et al., 2019; Wang et al., 2019), graph data purification (Entezari et al., 2020; Jin et al., 2020c; Wu et al., 2019a; Kipf & Welling, 2016), perturbation detection (Xu et al., 2018; Ioannidis et al., 2019; Jin et al., 2020b), and certified defense (Schuchardt et al., 2020; Wang et al., 2021; Bojchevski & Günnemann, 2019; Zügner & Günnemann, 2020; Jia et al., 2020). Adversarial training aims to inject adversarial examples (e.g., edges) during training, such that the GNN tends to yield correct predictions for adversarial examples during inference (Xu et al., 2019; Dai et al., 2019; Wang et al., 2019). Graph data purification also works during training, where graph data is purified during learning to weaken the influence of adversarial examples (Entezari et al., 2020; Jin et al., 2020c; Wu et al., 2019a; Kipf & Welling, 2016). Perturbation detection is mostly applied in the pre-processing stage, where adversarial edges or nodes can be identified before training (Xu et al., 2018; Ioannidis et al., 2019; Jin et al., 2020b). Different from them, certified defense is the only approach that secures GNNs theoretically, such that attackers cannot find any adversary to fool the GNNs (Schuchardt et al., 2020; Wang et al., 2021; Bojchevski & Günnemann, 2019; Zügner & Günnemann, 2020; Jia et al., 2020). Note that most certified defense approaches only secure the prediction for a specific data point (e.g., a node in node classification). Different from them, ELEGANT enables us to secure the fairness level for GNNs, which naturally entwines with all predictions in the test set.

## 6 CONCLUSION

In this paper, we study a novel problem of certifying GNN node classifiers on fairness. To address this problem, we propose ELEGANT, a framework designed to achieve certification on top of any optimized GNN node classifier associated with certain perturbation budgets, ensuring that it is impossible for attackers to degrade the fairness level of predictions within such budgets. Notably, ELEGANT is designed to serve as a plug-and-play framework for any optimized GNNs and does not rely on any assumptions regarding GNN structures or re-training processes. Extensive experiments verify the strong effectiveness and generalizability of ELEGANT across multiple GNN architectures and real-world datasets. While this paper primarily focuses on the widely studied node classification task, we also highlight the potential for extending this study to other graph-related tasks as a future research direction. We expect positive broader impacts including deploying fairness-safe GNNs in applications, and no significant negative broader impact needs to be highlighted here.

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

# Appendix

## Table of Contents

## A  PROOFS

For better clarity, for a matrix $X$, we use $X[i, j]$ to denote the element at the $i$-th row and the $j$-th column; for a vector $x$, we use $x[i]$ to denote its $i$-th component.

### A.1  PROOF OF THEOREM 1

To prove Theorem 1, we formulate the theoretical prerequisite that Theorem 1 relies on as Lemma A 1. Similarly, the proof of Lemma A 1 relies on the results in Lemma A 2, and the proof of Lemma A 2 is based on Lemma A 3.

*Proof.* For simplification, we reshape the matrix $X \in \mathbb{R}^{n \times d}$ to the vector $x \in \mathbb{R}^{nd}$. We denote $h_c(x) = \Pr(g(f_{\theta^*}, A, X + \gamma_X(\omega_X, \mathcal{V}_{\text{vul}}), \eta, \mathcal{V}_{\text{tst}}) = c)$ as the function that returns $P(c)$. Without loss of generality, we assume $\operatorname{argmax}_{c \in \{0,1\}} P(c) = 1$, so we have $\operatorname{argmin}_{c \in \{0,1\}} P(c) = 0$ consequently. We use $\Delta_X$ to denote a perturbation on $X$ that satisfies $\Delta_X \leq \tilde{\epsilon}_X$, and $\Delta_x$ to denote the reshaped vector of $\Delta_X$. Denote $\Phi^{-1}(\cdot)$ as the inverse of the standard Gaussian cumulative distribution function. According to Lemma A 1, we have $\Phi^{-1}(h_c(x))$ as a Lipschitz continuous function with a Lipschitz constant of $\frac{1}{\sigma}$ where $\sigma$ is the standard deviation of the Gaussian noise $\omega_X$. Based on the property of Lipschitz continuous functions, we have

$$|\Phi^{-1}(h_c(x + \Delta_x)) - \Phi^{-1}(h_c(x))| \leq \frac{\tilde{\epsilon}_X}{\sigma}.$$

Correspondingly, we have the following bounds for the output probabilities of class 0 and 1

$$\Phi^{-1}(h_1(x + \Delta_x)) - \Phi^{-1}(h_1(x)) \geq -\frac{\tilde{\epsilon}_X}{\sigma}, \tag{2}$$

$$\Phi^{-1}(h_0(x + \Delta_x)) - \Phi^{-1}(h_0(x)) \leq \frac{\tilde{\epsilon}_X}{\sigma}. \tag{3}$$

Combine Equation (2) and Equation (3), and we have

$$\Phi^{-1}(h_1(x + \Delta_x)) - \Phi^{-1}(h_0(x + \Delta_x)) \geq \Phi^{-1}(h_1(x)) - \Phi^{-1}(h_0(x)) - \frac{2\tilde{\epsilon}_X}{\sigma}. \tag{4}$$

Recall that $\tilde{\epsilon}_X = \frac{\sigma}{2}(\Phi^{-1}(\max_{c \in \{0,1\}} P(c)) - \Phi^{-1}(\min_{c \in \{0,1\}} P(c))) = \frac{\sigma}{2}(\Phi^{-1}(P(1)) - \Phi^{-1}(P(0))) = \frac{\sigma}{2}(\Phi^{-1}(h_1(x)) - \Phi^{-1}(h_0(x)))$, combine this condition with Equation (4), we have

$$\Phi^{-1}(h_1(x + \Delta_x)) - \Phi^{-1}(h_0(x + \Delta_x)) \geq 0. \tag{5}$$

Based on the strictly non-decreasing property of $\Phi(\cdot)$, we have

$$h_1(x + \Delta_x) \geq h_0(x + \Delta_x). \tag{6}$$

In Equation (6), $h_1(x + \Delta_x)$ and $h_0(x + \Delta_x)$ stand for the output probabilities of class 1 and 0 after the perturbation, correspondingly. Hence, the output for $\tilde{g}_X$ will not change after the perturbation (still class 1). Noting that the exact probabilities $\max_{c \in \{0,1\}} P(c)$ and $\min_{c \in \{0,1\}} P(c)$ are difficult to calculate in practice, we can use a tractable lower bound $\underline{p_{\max}}$ and upper bound $\overline{p_{\min}}$ such that $\max_{c \in \{0,1\}} P(c) \geq \underline{p_{\max}} \geq \overline{p_{\min}} \geq \min_{c \in \{0,1\}} P(c)$ to replace them in $\tilde{\epsilon}_X$ as $\tilde{\epsilon}_X = \frac{\sigma}{2}(\Phi^{-1}(\underline{p_{\max}}) - \Phi^{-1}(\overline{p_{\min}}))$. Because the practical perturbation budget $\tilde{\epsilon}_X$ derived by tractable bounds is smaller than the true budget, we can still obtain the same result as Equation (6). $\qquad\square$

**Lemma A 1.** *Denote $h_c(x) = \Pr(g(f_{\theta^*}, A, X + \gamma_X(\omega_X, \mathcal{V}_{vul}), \eta, \mathcal{V}_{tst}) = c)$ as the function that returns $P(c)$. Then, the function $\Phi^{-1}(h_c(x))$ is a Lipschitz continuous function with respect to $x$ with a Lipschitz constant $L_\Phi = \frac{1}{\sigma}$, where $\Phi^{-1}(\cdot)$ is the inverse of the standard Gaussian cumulative distribution function.*

*Proof.* To prove the Lipschitz continuity of $\Phi^{-1}(h_c(\boldsymbol{x}))$, we should find an upper bound of the norm of the gradient $\|\nabla_{\boldsymbol{x}}\Phi^{-1}(h_c(\boldsymbol{x}))\|_2$, denoted as $L_\Phi$. The gradient $\nabla_{\boldsymbol{x}}\Phi^{-1}(h_c(\boldsymbol{x}))$ is computed as

$$\nabla_{\boldsymbol{x}}\Phi^{-1}(h_c(\boldsymbol{x})) = \frac{\nabla_{\boldsymbol{x}}h_c(\boldsymbol{x})}{\Phi'(\Phi^{-1}(h_c(\boldsymbol{x})))}$$

$$= \sqrt{2\pi}\exp(\frac{1}{2}\Phi^{-1}(h_c(\boldsymbol{x}))^2)\nabla_{\boldsymbol{x}}h_c(\boldsymbol{x}).$$

Therefore, the norm $\|\nabla_{\boldsymbol{x}}\Phi^{-1}(h_c(\boldsymbol{x}))\|_2$ is computed as

$$\|\nabla_{\boldsymbol{x}}\Phi^{-1}(h_c(\boldsymbol{x}))\|_2 = \sqrt{2\pi}\exp(\frac{1}{2}\Phi^{-1}(h_c(\boldsymbol{x}))^2)\|\nabla_{\boldsymbol{x}}h_c(\boldsymbol{x})\|_2.$$

According to Lemma A 2, the upper bound of $\|\nabla_{\boldsymbol{x}}h_c(\boldsymbol{x})\|_2$ is $\frac{1}{\sqrt{2\pi}\sigma}\exp(-\frac{1}{2}\Phi^{-1}(h_c(\boldsymbol{x}))^2)$. Consequently, we have

$$\|\nabla_{\boldsymbol{x}}\Phi^{-1}(h_c(\boldsymbol{x}))\|_2 \leq \frac{1}{\sigma}.$$

Finally, we have obtained the Lipschitz constant of $\Phi^{-1}(h_c(\boldsymbol{x}))$ as $L_\Phi = \frac{1}{\sigma}$ and verified its Lipschitz continuity.

$\square$

**Lemma A 2.** *Denote $h_c(\boldsymbol{x}) = \Pr(g(f_{\boldsymbol{\theta}^*}, \boldsymbol{A}, \boldsymbol{X} + \gamma_{\boldsymbol{X}}(\boldsymbol{\omega}_{\boldsymbol{X}}, \mathcal{V}_{vul}), \eta, \mathcal{V}_{tst}) = c)$ as the function that returns $P(c)$. Then, the function $h_c(\boldsymbol{x})$ is a Lipschitz continuous function with respect to $\boldsymbol{x}$ with a Lipschitz constant $L_h = \frac{1}{\sqrt{2\pi}\sigma}\exp(-\frac{1}{2}\Phi^{-1}(h_c(\boldsymbol{x}))^2)$, where $\Phi^{-1}(\cdot)$ is the inverse of the standard Gaussian cumulative distribution function.*

*Proof.* To prove the Lipschitz continuity of $h_c(\boldsymbol{x})$, we should prove that the norm of the gradient $\|\nabla_{\boldsymbol{x}}h_c(\boldsymbol{x})\|_2$ is bounded by some constant $L_h$, i.e. $L_h = \sup_h\|\nabla_{\boldsymbol{x}}h_c(\boldsymbol{x})\|_2$. Let $\boldsymbol{\omega}_{vul} = \gamma_{\boldsymbol{X}}(\boldsymbol{\omega}_{\boldsymbol{X}}, \mathcal{V}_{vul}) \in \mathbb{R}^{nd}$ where $\boldsymbol{\omega}_{vul}[i] \sim \mathcal{N}(0, \sigma^2)$ when $i \in \mathcal{V}_{vul}$ and $\boldsymbol{\omega}_{vul}[i] = 0$ otherwise. Consequently, we have $h_c(\boldsymbol{x}) = \Pr(g(f_{\boldsymbol{\theta}^*}, \boldsymbol{A}, \boldsymbol{x} + \boldsymbol{\omega}_{vul}, \eta, \mathcal{V}_{tst}) = 1)$. Then, we compute the gradient $\nabla_{\boldsymbol{x}}h_c(\boldsymbol{x})$ as follows.

$$\nabla_{\boldsymbol{x}}h_c(\boldsymbol{x}) = \nabla_{\boldsymbol{x}}\Pr(g(f_{\boldsymbol{\theta}^*}, \boldsymbol{A}, \boldsymbol{x} + \boldsymbol{\omega}_{vul}, \eta, \mathcal{V}_{tst}) = c)$$

$$= \nabla_{\boldsymbol{x}}\mathbb{E}_{\boldsymbol{\omega}_{vul}}[g(f_{\boldsymbol{\theta}^*}, \boldsymbol{A}, \boldsymbol{x} + \boldsymbol{\omega}_{vul}, \eta, \mathcal{V}_{tst})]$$

$$= \nabla_{\boldsymbol{x}}\int_{\mathbb{R}^{|\mathcal{V}_{vul}|}} g(f_{\boldsymbol{\theta}^*}, \boldsymbol{A}, \boldsymbol{x} + \boldsymbol{\omega}_{vul}, \eta, \mathcal{V}_{tst})(2\pi\sigma^2)^{-\frac{|\mathcal{V}_{vul}|}{2}}\exp(-\frac{\|\boldsymbol{\omega}_{vul}\|_2^2}{2\sigma^2})d\boldsymbol{\omega}_{vul}.$$

Substituting $\boldsymbol{t} = \boldsymbol{x} + \boldsymbol{\omega}_{vul}$ into the above integration, we have

$$\nabla_{\boldsymbol{x}}h_c(\boldsymbol{x}) = \nabla_{\boldsymbol{x}}\int_{\mathbb{R}^{|\mathcal{V}_{vul}|}} g(f_{\boldsymbol{\theta}^*}, \boldsymbol{A}, \boldsymbol{t}, \eta, \mathcal{V}_{tst})(2\pi\sigma^2)^{-\frac{|\mathcal{V}_{vul}|}{2}}\exp(-\frac{\|\boldsymbol{t} - \boldsymbol{x}\|_2^2}{2\sigma^2})d\boldsymbol{t}$$

$$= \int_{\mathbb{R}^{|\mathcal{V}_{vul}|}} g(f_{\boldsymbol{\theta}^*}, \boldsymbol{A}, \boldsymbol{t}, \eta, \mathcal{V}_{tst})(2\pi\sigma^2)^{-\frac{|\mathcal{V}_{vul}|}{2}}\nabla_{\boldsymbol{x}}\exp(-\frac{\|\boldsymbol{t} - \boldsymbol{x}\|_2^2}{2\sigma^2})d\boldsymbol{t}$$

$$= \int_{\mathbb{R}^{|\mathcal{V}_{vul}|}} g(f_{\boldsymbol{\theta}^*}, \boldsymbol{A}, \boldsymbol{t}, \eta, \mathcal{V}_{tst})(2\pi\sigma^2)^{-\frac{|\mathcal{V}_{vul}|}{2}}\exp(-\frac{\|\boldsymbol{t} - \boldsymbol{x}\|_2^2}{2\sigma^2})\frac{\boldsymbol{t} - \boldsymbol{x}}{\sigma^2}d\boldsymbol{t}$$

$$= \int_{\mathbb{R}^{|\mathcal{V}_{vul}|}} g(f_{\boldsymbol{\theta}^*}, \boldsymbol{A}, \boldsymbol{x} + \boldsymbol{\omega}_{vul}, \eta, \mathcal{V}_{tst})(2\pi\sigma^2)^{-\frac{|\mathcal{V}_{vul}|}{2}}\exp(-\frac{\|\boldsymbol{\omega}_{vul}\|_2^2}{2\sigma^2})\frac{\boldsymbol{\omega}_{vul}}{\sigma^2}d\boldsymbol{\omega}_{vul}$$

$$= \frac{1}{\sigma^2}\mathbb{E}_{\boldsymbol{\omega}_{vul}}[\boldsymbol{\omega}_{vul}g(f_{\boldsymbol{\theta}^*}, \boldsymbol{A}, \boldsymbol{x} + \boldsymbol{\omega}_{vul}, \eta, \mathcal{V}_{tst})]$$

$$= \frac{1}{\sigma}\mathbb{E}_{\boldsymbol{\omega}'_{vul}}[\boldsymbol{\omega}'_{vul}g(f_{\boldsymbol{\theta}^*}, \boldsymbol{A}, \boldsymbol{x} + \sigma\boldsymbol{\omega}'_{vul}, \eta, \mathcal{V}_{tst})].$$

Here, $\boldsymbol{\omega}'_{vul}$ is a normalized random vector that $\boldsymbol{\omega}'_{vul}[i] \sim \mathcal{N}(0, 1)$ when $i \in \mathcal{V}_{vul}$ and $\boldsymbol{\omega}'_{vul}[i] = 0$ otherwise. Next, we compute the norm of the gradient $\|\nabla_{\boldsymbol{x}}h_c(\boldsymbol{x})\|_2$ as

$$\|\nabla_{\boldsymbol{x}}h_c(\boldsymbol{x})\|_2 = \sup_{\|\boldsymbol{v}\|_2=1}\boldsymbol{v}^\top\nabla_{\boldsymbol{x}}h_c(\boldsymbol{x})$$

$$= \frac{1}{\sigma}\sup_{\|\boldsymbol{v}\|_2=1}\mathbb{E}_{\boldsymbol{\omega}'_{vul}}[\boldsymbol{v}^\top\boldsymbol{\omega}'_{vul}g(f_{\boldsymbol{\theta}^*}, \boldsymbol{A}, \boldsymbol{x} + \sigma\boldsymbol{\omega}'_{vul}, \eta, \mathcal{V}_{tst})]. \tag{7}$$

To find $L_h$, we should consider the worst case (with the largest Lipschitz constant) among all possible classifiers. We let $\tilde{g}(\boldsymbol{A}, \boldsymbol{\omega}'_{\text{vul}}) = g(f_{\boldsymbol{\theta}^*}, \boldsymbol{A}, \boldsymbol{x} + \sigma\boldsymbol{\omega}'_{\text{vul}}, \eta, \mathcal{V}_{\text{tst}})$. Then, we have the following optimization problem for solving $L_h$

$$\sup_{\tilde{g}} \mathbb{E}_{\boldsymbol{\omega}'_{\text{vul}}}[\boldsymbol{v}^\top\boldsymbol{\omega}'_{\text{vul}}\tilde{g}(\boldsymbol{A}, \boldsymbol{\omega}'_{\text{vul}})]$$

$$s.t. \quad \tilde{g}(\boldsymbol{A}, \boldsymbol{\omega}'_{\text{vul}}) \in [0, 1], \ \|\boldsymbol{v}\|_2 = 1, \mathbb{E}_{\boldsymbol{\omega}'_{\text{vul}}}[\tilde{g}(\boldsymbol{A}, \boldsymbol{\omega}'_{\text{vul}})] = h_c(\boldsymbol{x}). \tag{8}$$

The rationale of this optimization problem is that we aim to find the function with the largest Lipschitz constant (objective) among all possible classifiers $\tilde{g}$ with the same smoothing output $h_c(\boldsymbol{x})$ (constraints) when fixing the variable $\boldsymbol{v}$. After solving this problem, we can find the largest objective among all possible $\boldsymbol{v}$ as $L_h$. To solve this problem, we have the following lemma.

Based on Lemma A 3, we have $\tilde{g}^*(\boldsymbol{A}, \boldsymbol{\omega}'_{\text{vul}}) = \mathbb{1}(\boldsymbol{v}^\top\boldsymbol{\omega}'_{\text{vul}} \geq -\varepsilon_{\boldsymbol{v}}\Phi^{-1}(h_c(\boldsymbol{x})))$ as the solution of the optimization problem in Equation (8). Next, we can compute the maximal objective when fixing the variable $\boldsymbol{v}$ as

$$\mathbb{E}_{\boldsymbol{\omega}'_{\text{vul}}}[\boldsymbol{v}^\top\boldsymbol{\omega}'_{\text{vul}} \cdot \mathbb{1}(\boldsymbol{v}^\top\boldsymbol{\omega}'_{\text{vul}} \geq -\varepsilon_{\boldsymbol{v}}\Phi^{-1}(h_c(\boldsymbol{x})))]$$

$$= \mathbb{E}_{\omega \sim \mathcal{N}(0, \varepsilon_{\boldsymbol{v}}^2)}[\omega \cdot \mathbb{1}(\omega \geq -\varepsilon_{\boldsymbol{v}}\Phi^{-1}(h_c(\boldsymbol{x})))]$$

$$= \mathbb{E}_{\omega' \sim \mathcal{N}(0,1)}[\varepsilon_{\boldsymbol{v}}\omega' \cdot \mathbb{1}(\omega' \geq -\Phi^{-1}(h_c(\boldsymbol{x})))] \text{ (let } \omega = \varepsilon_{\boldsymbol{v}}\omega') \tag{9}$$

$$= \frac{\varepsilon_{\boldsymbol{v}}}{\sqrt{2\pi}} \int_{-\Phi^{-1}(h_c(\boldsymbol{x}))}^{+\infty} \omega\exp(-\frac{\omega}{2})d\omega$$

$$= \frac{\varepsilon_{\boldsymbol{v}}}{\sqrt{2\pi}}\exp(-\frac{1}{2}\Phi^{-1}(h_c(\boldsymbol{x}))^2).$$

Therefore, we have $\sup_{\tilde{g}} \mathbb{E}_{\boldsymbol{\omega}'_{\text{vul}}}[\boldsymbol{v}^\top\boldsymbol{\omega}'_{\text{vul}}\tilde{g}(\boldsymbol{A}, \boldsymbol{\omega}'_{\text{vul}})] = \frac{\varepsilon_{\boldsymbol{v}}}{\sqrt{2\pi}}\exp(-\frac{1}{2}\Phi^{-1}(h_c(\boldsymbol{x}))^2)$. Combining this result with Equation (7), we have

$$L_h = \sup_h \|\nabla_{\boldsymbol{x}} h_c(\boldsymbol{x})\|_2$$

$$= \sup_{\tilde{g}, \|\boldsymbol{v}\|_2=1} \frac{1}{\sigma}\mathbb{E}_{\boldsymbol{\omega}'_{\text{vul}}}[\boldsymbol{v}^\top\boldsymbol{\omega}'_{\text{vul}}\tilde{g}(\boldsymbol{A}, \boldsymbol{\omega}'_{\text{vul}})]$$

$$= \sup_{\|\boldsymbol{v}\|_2=1} \frac{\varepsilon_{\boldsymbol{v}}}{\sqrt{2\pi}\sigma}\exp(-\frac{1}{2}\Phi^{-1}(h_c(\boldsymbol{x}))^2)$$

$$= \frac{1}{\sqrt{2\pi}\sigma}\exp(-\frac{1}{2}\Phi^{-1}(h_c(\boldsymbol{x}))^2).$$

Finally, we have proved that the function $h_c(\boldsymbol{x}) = \Pr(g(f_{\boldsymbol{\theta}^*}, \boldsymbol{A}, \boldsymbol{X} + \gamma_{\boldsymbol{X}}(\boldsymbol{\omega}_{\boldsymbol{X}}, \mathcal{V}_{\text{vul}}), \eta, \mathcal{V}_{\text{tst}}) = c)$ is a Lipschitz continuous function with respect to variable $\boldsymbol{x}$. $\qquad\square$

**Lemma A 3.** *The solution to the optimization problem in Equation (8) is $\tilde{g}^*(\boldsymbol{A}, \boldsymbol{\omega}'_{vul}) = \mathbb{1}(\boldsymbol{v}^\top\boldsymbol{\omega}'_{vul} \geq -\varepsilon_{\boldsymbol{v}}\Phi^{-1}(h_c(\boldsymbol{x})))$, where $\varepsilon_{\boldsymbol{v}}^2 = \sum_{i \in \mathcal{V}_{vul}} \boldsymbol{v}[i]^2$.*

*Proof.* First, we clarify the rationale for solving this problem. We note that $\boldsymbol{v}^\top\boldsymbol{\omega}'_{\text{vul}} \sim \mathcal{N}(0, \varepsilon_{\boldsymbol{v}}^2)$ (based on the property of independent and identically distributed Gaussian), this optimization problem can be regarded as the reweighting of a Gaussian distribution where the range of the weight function $\tilde{g}(\boldsymbol{A}, \boldsymbol{\omega}'_{\text{vul}})$ is $[0, 1]$ and the constraint of the weight function is given by $\mathbb{E}_{\boldsymbol{\omega}'_{\text{vul}}}[\tilde{g}(\boldsymbol{A}, \boldsymbol{\omega}'_{\text{vul}})] = h_c(\boldsymbol{x})$. A straightforward solution here is to let the weight function at a large value of $\boldsymbol{v}^\top\boldsymbol{\omega}'_{\text{vul}}$ as large as possible. We let $\tilde{g}(\boldsymbol{A}, \boldsymbol{\omega}'_{\text{vul}}) = 1$ where $\boldsymbol{v}^\top\boldsymbol{\omega}'_{\text{vul}} \geq -\varepsilon_{\boldsymbol{v}}\Phi^{-1}(h_c(\boldsymbol{x}))(\boldsymbol{\omega}'_{\text{vul}})$ and $\tilde{g}(\boldsymbol{A}, \boldsymbol{\omega}'_{\text{vul}}) = 0$ otherwise. Here $\Phi(\cdot)$ is the cumulative distribution function of $\mathcal{N}(0, 1)$.

Next, we prove that $\tilde{g}^*(\boldsymbol{A}, \boldsymbol{\omega}'_{\text{vul}}) = \mathbb{1}(\boldsymbol{v}^\top\boldsymbol{\omega}'_{\text{vul}} \geq -\varepsilon_{\boldsymbol{v}}\Phi^{-1}(h_c(\boldsymbol{x})))$ is the exact solution of the optimization problem in Equation (8). We first verify that $\tilde{g}^*$ is a solution. It is obvious that $\tilde{g}^*$ suffices the first two constraints because the range of the indicator function is $\{0, 1\}$. For the last constraint, $\mathbb{E}_{\boldsymbol{\omega}'_{\text{vul}}}[\mathbb{1}(\boldsymbol{v}^\top\boldsymbol{\omega}'_{\text{vul}} \geq -\varepsilon_{\boldsymbol{v}}\Phi^{-1}(h_c(\boldsymbol{x})))]$ is actually the probability of $\boldsymbol{v}^\top\boldsymbol{\omega}'_{\text{vul}}$ being larger than $-\varepsilon_{\boldsymbol{v}}\Phi^{-1}(h_c(\boldsymbol{x}))$, which equals to $(h_c(\boldsymbol{x})$ apparently because $\boldsymbol{v}^\top\boldsymbol{\omega}'_{\text{vul}}/\varepsilon_{\boldsymbol{v}} \sim \mathcal{N}(0, 1)$. Therefore, $\tilde{g}^*$ satisfies all three constraints. We then prove that $\tilde{g}^*$ is the optimal solution. We assume $\tilde{g} \neq \tilde{g}^*$ is

another classifier that also suffices the constraints in the optimization problem in Equation (8). We use $\mathcal{S}$ to denote the support set $\{s \mid \tilde{g}^*(\boldsymbol{A}, \boldsymbol{\omega}'_{\text{vul}}) \neq 0\}$. Based on the final constraint in the optimization problem in Equation (8), we have

$$\mathbb{E}_{\boldsymbol{\omega}'_{\text{vul}}}[\tilde{g}^*(\boldsymbol{A}, \boldsymbol{\omega}'_{\text{vul}}) - \tilde{g}(\boldsymbol{A}, \boldsymbol{\omega}'_{\text{vul}})] = 0.$$

We divide this equation into two parts as

$$\mathbb{E}_{\boldsymbol{\omega}'_{\text{vul}}}[(\tilde{g}^*(\boldsymbol{A}, \boldsymbol{\omega}'_{\text{vul}}) - \tilde{g}(\boldsymbol{A}, \boldsymbol{\omega}'_{\text{vul}}))\mathbb{1}(\boldsymbol{\omega}'_{\text{vul}} \in \mathcal{S})] + \mathbb{E}_{\boldsymbol{\omega}'_{\text{vul}}}[(\tilde{g}^*(\boldsymbol{A}, \boldsymbol{\omega}'_{\text{vul}}) - \tilde{g}(\boldsymbol{A}, \boldsymbol{\omega}'_{\text{vul}}))\mathbb{1}(\boldsymbol{\omega}'_{\text{vul}} \in \mathcal{S}^{\complement})] = 0, \tag{10}$$

where $\mathcal{S}^{\complement}$ denotes the complement set of $\mathcal{S}$. We know that $\tilde{g}^*(\boldsymbol{A}, \boldsymbol{\omega}'_{\text{vul}}) \equiv 1$ for $\boldsymbol{\omega}'_{\text{vul}} \in \mathcal{S}$. We also know that $\tilde{g}(\boldsymbol{A}, \boldsymbol{\omega}'_{\text{vul}}) \leq 1$ and $\tilde{g}(\boldsymbol{A}, \boldsymbol{\omega}'_{\text{vul}})$ cannot always equal to 1 for $\boldsymbol{\omega}'_{\text{vul}} \in \mathcal{S}$ because $\tilde{g} \neq \tilde{g}^*$. Therefore, we have

$$\mathbb{E}_{\boldsymbol{\omega}'_{\text{vul}}}[(\tilde{g}^*(\boldsymbol{A}, \boldsymbol{\omega}'_{\text{vul}}) - \tilde{g}(\boldsymbol{A}, \boldsymbol{\omega}'_{\text{vul}}))\mathbb{1}(\boldsymbol{\omega}'_{\text{vul}} \in \mathcal{S})] > 0,$$
$$\mathbb{E}_{\boldsymbol{\omega}'_{\text{vul}}}[(\tilde{g}^*(\boldsymbol{A}, \boldsymbol{\omega}'_{\text{vul}}) - \tilde{g}(\boldsymbol{A}, \boldsymbol{\omega}'_{\text{vul}}))\mathbb{1}(\boldsymbol{\omega}'_{\text{vul}} \in \mathcal{S}^{\complement})] < 0. \tag{11}$$

Moreover, we have $\boldsymbol{v}^\top \boldsymbol{\omega}_1 > \boldsymbol{v}^\top \boldsymbol{\omega}_0$ for any $\boldsymbol{\omega}_1 \in \mathcal{S}$ and $\boldsymbol{\omega}_0 \in \mathcal{S}^{\complement}$. Finally, combine this result with Equation (10) and Equation (11), we have

$$\mathbb{E}_{\boldsymbol{\omega}'_{\text{vul}}}[\boldsymbol{v}^\top \boldsymbol{\omega}'_{\text{vul}}(\tilde{g}^*(\boldsymbol{A}, \boldsymbol{\omega}'_{\text{vul}}) - \tilde{g}(\boldsymbol{A}, \boldsymbol{\omega}'_{\text{vul}}))\mathbb{1}(\boldsymbol{\omega}'_{\text{vul}} \in \mathcal{S})]$$
$$+ \mathbb{E}_{\boldsymbol{\omega}'_{\text{vul}}}[\boldsymbol{v}^\top \boldsymbol{\omega}'_{\text{vul}}(\tilde{g}^*(\boldsymbol{A}, \boldsymbol{\omega}'_{\text{vul}}) - \tilde{g}(\boldsymbol{A}, \boldsymbol{\omega}'_{\text{vul}}))\mathbb{1}(\boldsymbol{\omega}'_{\text{vul}} \in \mathcal{S}^{\complement})] > 0.$$

Consequently, we have $\mathbb{E}_{\boldsymbol{\omega}'_{\text{vul}}}[\boldsymbol{v}^\top \boldsymbol{\omega}'_{\text{vul}} \tilde{g}^*(\boldsymbol{A}, \boldsymbol{\omega}'_{\text{vul}})] - \mathbb{E}_{\boldsymbol{\omega}'_{\text{vul}}}[\boldsymbol{v}^\top \boldsymbol{\omega}'_{\text{vul}} \tilde{g}(\boldsymbol{A}, \boldsymbol{\omega}'_{\text{vul}})] > 0$. Therefore, we have proved that $\tilde{g}^*(\boldsymbol{A}, \boldsymbol{\omega}'_{\text{vul}}) = \mathbb{1}(\boldsymbol{v}^\top \boldsymbol{\omega}'_{\text{vul}} \geq -\varepsilon_{\boldsymbol{v}} \Phi^{-1}(h_c(\boldsymbol{x})))$ is the exact optimal solution of the optimization problem in Equation (8). $\qquad\square$

## A.2 PROOF OF LEMMA 1

*Proof.* The tractable perturbation budgets $\epsilon_{\boldsymbol{A}}$ and $\epsilon_{\boldsymbol{X}}$ can be obtained according to Theorem 2 and Theorem 4, correspondingly. $\qquad\square$

## A.3 PROOF OF LEMMA 2

*Proof.* The tractable probability lower bound $\underline{P_{\tilde{g}_{\boldsymbol{X}}=1}}$ can be obtained according to Theorem 3. $\qquad\square$

## A.4 PROOF OF THEOREM 2

*Proof.* To certify the fairness level, we assume that $\tilde{g}_{\boldsymbol{A}, \boldsymbol{X}}(\boldsymbol{A}, \boldsymbol{X}) = 1$. Refer to Lemma 2, we have $\Pr(\tilde{g}_{\boldsymbol{X}}(\boldsymbol{A} \oplus \boldsymbol{\Delta}_{\boldsymbol{A}} \oplus \boldsymbol{\Gamma}_{\boldsymbol{A}}, \boldsymbol{X}) = 1) \geq \underline{P_{\tilde{g}_{\boldsymbol{X}}=1}}$. For any structure perturbation $\|\boldsymbol{\Delta}_{\boldsymbol{A}}\|_0 \leq \tilde{\epsilon}_{\boldsymbol{A}}$, we combine this result with Equation (1) and obtain that

$$\Pr(\tilde{g}_{\boldsymbol{X}}(\boldsymbol{A} \oplus \boldsymbol{\Delta}_{\boldsymbol{A}} \oplus \boldsymbol{\Gamma}_{\boldsymbol{A}}, \boldsymbol{X}) = 1) \geq \underline{P_{\tilde{g}_{\boldsymbol{X}}=1}} > 0.5, \tag{12}$$

As a consequence, we have $\tilde{g}_{\boldsymbol{A}, \boldsymbol{X}}(\boldsymbol{A} \oplus \boldsymbol{\Delta}_{\boldsymbol{A}}, \boldsymbol{X}) = \tilde{g}_{\boldsymbol{A}, \boldsymbol{X}}(\boldsymbol{A}, \boldsymbol{X})$ for any structure perturbation $\|\boldsymbol{\Delta}_{\boldsymbol{A}}\|_0 \leq \tilde{\epsilon}_{\boldsymbol{A}}$. $\qquad\square$

## A.5 PROOF OF THEOREM 3

*Proof.* To prove Theorem 3, we formulate the theoretical prerequisite that Theorem 3 relies on as Lemma A 4. The following Lemma A 4 indicates the relation between $\boldsymbol{A} \oplus \boldsymbol{\Gamma}_{\boldsymbol{A}}$ and $\boldsymbol{A} \oplus \boldsymbol{\Delta}_{\boldsymbol{A}} \oplus \boldsymbol{\Gamma}_{\boldsymbol{A}}$.

**Lemma A 4.** *Let $\boldsymbol{X}$ and $\boldsymbol{Y}$ be two random vectors in the discrete space $\{0,1\}^n$ with prabability distributions $\Pr(\boldsymbol{X})$ and $\Pr(\boldsymbol{Y})$, correspondingly. Let $h : \{0,1\}^n \to \{0,1\}$ be a random or deterministic function. Let $\mathcal{S}_1 = \{\boldsymbol{z} \in \{0,1\}^n : \frac{\Pr(\boldsymbol{X}=\boldsymbol{z})}{\Pr(\boldsymbol{Y}=\boldsymbol{z})} > r\}$ and $\mathcal{S}_2 = \{\boldsymbol{z} \in \{0,1\}^n : \frac{\Pr(\boldsymbol{X}=\boldsymbol{z})}{\Pr(\boldsymbol{Y}=\boldsymbol{z})} = r\}$ for some $r > 0$. Assume $\mathcal{S}_3 \subseteq \mathcal{S}_2$ and $\mathcal{S} = \mathcal{S}_1 \cup \mathcal{S}_3$. If $\Pr(h(\boldsymbol{X}) = 1) \geq \Pr(\boldsymbol{X} \in \mathcal{S})$, then $\Pr(h(\boldsymbol{Y}) = 1) \geq \Pr(\boldsymbol{Y} \in \mathcal{S})$.*

*Proof.* Note that we have $\frac{\Pr(\boldsymbol{X}=\boldsymbol{z})}{\Pr(\boldsymbol{Y}=\boldsymbol{z})} \geq r$ for any $\boldsymbol{z} \in \mathcal{S}$ and $\frac{\Pr(\boldsymbol{X}=\boldsymbol{z})}{\Pr(\boldsymbol{Y}=\boldsymbol{z})} \leq r$ for any $\boldsymbol{z} \in \mathcal{S}^{\complement}$. Assuming $h$ is random, we have

$$\Pr(h(\boldsymbol{Y}) = 1) - \Pr(\boldsymbol{Y} \in \mathcal{S})$$

$$= \sum_{\boldsymbol{z} \in \{0,1\}^n} \Pr(h(\boldsymbol{z}) = 1)\Pr(\boldsymbol{Y} = \boldsymbol{z}) - \sum_{\boldsymbol{z} \in \mathcal{S}} \Pr(\boldsymbol{Y} = \boldsymbol{z})$$

$$= \sum_{\boldsymbol{z} \in \mathcal{S}} \Pr(h(\boldsymbol{z}) = 1)\Pr(\boldsymbol{Y} = \boldsymbol{z}) + \sum_{\boldsymbol{z} \in \mathcal{S}^{\complement}} \Pr(h(\boldsymbol{z}) = 1)\Pr(\boldsymbol{Y} = \boldsymbol{z})$$

$$- \sum_{\boldsymbol{z} \in \mathcal{S}} \Pr(h(\boldsymbol{z}) = 1)\Pr(\boldsymbol{Y} = \boldsymbol{z}) - \sum_{\boldsymbol{z} \in \mathcal{S}} \Pr(h(\boldsymbol{z}) = 0)\Pr(\boldsymbol{Y} = \boldsymbol{z})$$

$$= \sum_{\boldsymbol{z} \in \mathcal{S}^{\complement}} \Pr(h(\boldsymbol{z}) = 1)\Pr(\boldsymbol{Y} = \boldsymbol{z}) - \sum_{\boldsymbol{z} \in \mathcal{S}} \Pr(h(\boldsymbol{z}) = 0)\Pr(\boldsymbol{Y} = \boldsymbol{z})$$

$$\geq \frac{1}{r}\left( \sum_{\boldsymbol{z} \in \mathcal{S}^{\complement}} \Pr(h(\boldsymbol{z}) = 1)\Pr(\boldsymbol{X} = \boldsymbol{z}) - \sum_{\boldsymbol{z} \in \mathcal{S}} \Pr(h(\boldsymbol{z}) = 0)\Pr(\boldsymbol{X} = \boldsymbol{z}) \right)$$

$$= \frac{1}{r}\left( \sum_{\boldsymbol{z} \in \mathcal{S}^{\complement}} \Pr(h(\boldsymbol{z}) = 1)\Pr(\boldsymbol{X} = \boldsymbol{z}) + \sum_{\boldsymbol{z} \in \mathcal{S}} \Pr(h(\boldsymbol{z}) = 1)\Pr(\boldsymbol{X} = \boldsymbol{z}) \right.$$

$$\left. - \sum_{\boldsymbol{z} \in \mathcal{S}} \Pr(h(\boldsymbol{z}) = 0)\Pr(\boldsymbol{X} = \boldsymbol{z}) - \sum_{\boldsymbol{z} \in \mathcal{S}} \Pr(h(\boldsymbol{z}) = 1)\Pr(\boldsymbol{X} = \boldsymbol{z}) \right)$$

$$= \frac{1}{r}\left( \sum_{\boldsymbol{z} \in \{0,1\}^n} \Pr(h(\boldsymbol{z}) = 1)\Pr(\boldsymbol{X} = \boldsymbol{z}) - \sum_{\boldsymbol{z} \in \mathcal{S}} \Pr(\boldsymbol{X} = \boldsymbol{z}) \right)$$

$$= \frac{1}{r}(\Pr(h(\boldsymbol{X}) = 1) - \Pr(\boldsymbol{X} \in \mathcal{S}))$$

$$\geq 0.$$

$\square$

Based on the definition of $\mathcal{H}$, we have $\Pr(\tilde{g}_{\boldsymbol{X}}(\boldsymbol{A} \oplus \boldsymbol{\Delta_A}, \boldsymbol{X}) = 1) \geq \Pr(\boldsymbol{A} \oplus \boldsymbol{\Delta_A} \in \mathcal{H})$ distinctly. According to Lemma A 4 (let $\boldsymbol{Y} = \boldsymbol{A} \oplus \boldsymbol{\Delta_A} \oplus \boldsymbol{\Gamma_A}$, $\boldsymbol{X} = \boldsymbol{A} \oplus \boldsymbol{\Delta_A}$, $\mathcal{S} = \mathcal{H}$, and $h = \tilde{g}_{\boldsymbol{X}}$), we have $\Pr(\tilde{g}_{\boldsymbol{X}}(\boldsymbol{A} \oplus \boldsymbol{\Delta_A} \oplus \boldsymbol{\Gamma_A}, \boldsymbol{X}) = 1) \geq \Pr(\boldsymbol{A} \oplus \boldsymbol{\Delta_A} \oplus \boldsymbol{\Gamma_A} \in \mathcal{H})$ ($\boldsymbol{X}$ can be seen as a constant here), correspondingly. Noting that the exact probability $\Pr(\tilde{g}_{\boldsymbol{X}}(\boldsymbol{A} \oplus \boldsymbol{\Gamma_A}, \boldsymbol{X}) = 1)$ is difficult to calculate, we use a practical lower bound $\underline{P_{\tilde{g}_{\boldsymbol{X}}=1}}^* \leq \Pr(\tilde{g}_{\boldsymbol{X}}(\boldsymbol{A} \oplus \boldsymbol{\Gamma_A}, \boldsymbol{X}) = 1)$ to replace it in practice. Because we also have $\Pr(\tilde{g}_{\boldsymbol{X}}(\boldsymbol{A} \oplus \boldsymbol{\Delta_A}, \boldsymbol{X}) = 1) \geq \Pr(\boldsymbol{A} \oplus \boldsymbol{\Delta_A} \in \mathcal{H})$ for $\mathcal{H}$ derived by $\underline{P_{\tilde{g}_{\boldsymbol{X}}=1}}^*$, the proof still holds.

$\square$

## A.6 SOLVING THE OPTIMIZATION PROBLEM IN THEOREM 2

We can compute $\Pr(\boldsymbol{A} \oplus \boldsymbol{\Delta_A} \oplus \boldsymbol{\Gamma_A} \in \mathcal{H})$ as

$$\Pr(\boldsymbol{A} \oplus \boldsymbol{\Delta_A} \oplus \boldsymbol{\Gamma_A} \in \mathcal{H}) = \sum_{j=\mu+1}^{n|\mathcal{V}_{\text{vul}}|} \Pr(\boldsymbol{A} \oplus \boldsymbol{\Delta_A} \oplus \boldsymbol{\Gamma_A} \in \mathcal{H}_j)$$

$$+ (\underline{P_{\tilde{g}_{\boldsymbol{X}}=1}}^* - \sum_{j=\mu+1}^{n|\mathcal{V}_{\text{vul}}|} \Pr(\boldsymbol{A} \oplus \boldsymbol{\Gamma_A} \in \mathcal{H}_j))/(\frac{\beta}{1-\beta})^{\mu}. \tag{13}$$

To compute Equation (13), we calculate the probability $\Pr(\boldsymbol{A} \oplus \boldsymbol{\Gamma_A} \in \mathcal{H}_m)$ and $\Pr(\boldsymbol{A} \oplus \boldsymbol{\Delta_A} \oplus \boldsymbol{\Gamma_A} \in \mathcal{H}_m)$ as

$$\Pr(\boldsymbol{A} \oplus \boldsymbol{\Gamma_A} \in \mathcal{H}_m) = \sum_{j=\max\{0,m\}}^{\min\{n|\mathcal{V}_{\text{vul}}|,n|\mathcal{V}_{\text{vul}}|+m\}} \beta_{n|\mathcal{V}_{\text{vul}}|-(j-m)}(1-\beta)^{(j-m)} t(m,j), \qquad (14)$$

$$\Pr(\boldsymbol{A} \oplus \boldsymbol{\Delta_A} \oplus \boldsymbol{\Gamma_A} \in \mathcal{H}_m) = \sum_{j=\max\{0,m\}}^{\min\{n|\mathcal{V}_{\text{vul}}|,n|\mathcal{V}_{\text{vul}}|+m\}} \beta_{n|\mathcal{V}_{\text{vul}}|-j}(1-\beta)^j t(m,j), \qquad (15)$$

where $-n|\mathcal{V}_{\text{vul}}| \le m \le n|\mathcal{V}_{\text{vul}}|$, $\|\boldsymbol{\Delta_A}\|_0 \le k$, and $t(m,j)$ is defined as

$$t(m,j) = \begin{cases} 0, & \text{if } (m+k) \bmod 2 \ne 0, \\ 0, & \text{if } 2j-m < k, \\ \binom{n|\mathcal{V}_{\text{vul}}|-k}{\frac{2j-m-k}{2}}\binom{k}{\frac{k-m}{2}}, & \text{otherwise.} \end{cases} \qquad (16)$$

With Equation (13), we can traverse the perturbation budget $\|\boldsymbol{\Delta_A}\|_0$ over $1, 2, \dots$ until $\Pr(\boldsymbol{A} \oplus \boldsymbol{\Delta_A} \oplus \boldsymbol{\Gamma_A} \in \mathcal{H}) < 0.5$.

### A.7 Proof of Theorem 4

*Proof.* Recall that $\tilde{g}_{\boldsymbol{A},\boldsymbol{X}}(\boldsymbol{A}, \boldsymbol{X}) = \arg\max_{c \in \{0,1\}} \Pr(\tilde{g}_{\boldsymbol{X}}(\boldsymbol{A} \oplus \boldsymbol{\Gamma_A}, \boldsymbol{X}) = c)$ and $\tilde{g}_{\boldsymbol{X}}(\boldsymbol{A}, \boldsymbol{X}) = \arg\max_{c \in \{0,1\}} \Pr(g(f_{\boldsymbol{\theta}^*}, \boldsymbol{A}, \boldsymbol{X} + \boldsymbol{\Gamma_X}, \eta, \mathcal{V}_{\text{tst}}) = c)$. To certify the fairness level, we assume that $\tilde{g}_{\boldsymbol{A},\boldsymbol{X}}(\boldsymbol{A}, \boldsymbol{X}) = 1$, which means that

$$\Pr(\tilde{g}_{\boldsymbol{X}}(\boldsymbol{A} \oplus \boldsymbol{\Gamma_A}, \boldsymbol{X}) = 1) > 0.5. \qquad (17)$$

For any perturbation $\|\boldsymbol{\Delta_X}\|_2 \le \epsilon_{\boldsymbol{X}} \le \epsilon_{\tilde{\boldsymbol{X}}}$, we have $\tilde{g}_{\boldsymbol{X}}(\boldsymbol{A} \oplus \boldsymbol{\Gamma_A}, \boldsymbol{X} + \boldsymbol{\Delta_X}) = \tilde{g}_{\boldsymbol{X}}(\boldsymbol{A} \oplus \boldsymbol{\Gamma_A}, \boldsymbol{X})$ for any $\boldsymbol{\Gamma_A} \in \bar{\mathcal{A}}$ where $\epsilon_{\tilde{\boldsymbol{X}}}$ is derived with classifier $\tilde{g}_{\boldsymbol{X}}(\boldsymbol{A} \oplus \boldsymbol{\Gamma_A}, \boldsymbol{X})$. Regarding the randomness of $\boldsymbol{\Gamma_A}$, we have

$$\Pr(\tilde{g}_{\boldsymbol{X}}(\boldsymbol{A} \oplus \boldsymbol{\Gamma_A}, \boldsymbol{X} + \boldsymbol{\Delta_X}) = 1) = \Pr(\tilde{g}_{\boldsymbol{X}}(\boldsymbol{A} \oplus \boldsymbol{\Gamma_A}, \boldsymbol{X}) = 1) > 0.5. \qquad (18)$$

Hence we obtain that $\tilde{g}_{\boldsymbol{A},\boldsymbol{X}}(\boldsymbol{A}, \boldsymbol{X} + \boldsymbol{\Delta_X}) = \tilde{g}_{\boldsymbol{A},\boldsymbol{X}}(\boldsymbol{A}, \boldsymbol{X})$ for any perturbation $\|\boldsymbol{\Delta_X}\|_2 \le \epsilon_{\boldsymbol{X}}$. $\square$

### A.8 Proof of Proposition 1

*Proof.* For the practical certification, we add perturbations within certified budgets and derive independent identically distributed output samples $\hat{\mathcal{Y}}'$ by Monte Carlo. For each sample $\hat{\boldsymbol{Y}}' \in \hat{\mathcal{Y}}'$, we have $\Pr(\pi(\hat{\boldsymbol{Y}}', \mathcal{V}_{\text{tst}}) > \eta) < 0.5$ according to Theorem 4 and Theorem 2. $\pi(\hat{\boldsymbol{Y}}, \mathcal{V}_{\text{tst}}) > \eta$ indicates that $\pi(\hat{\boldsymbol{Y}}', \mathcal{V}_{\text{tst}}) > \eta$ for any $\hat{\boldsymbol{Y}}' \in \hat{\mathcal{Y}}'$. Consequently, we have

$$\Pr(\pi(\hat{\boldsymbol{Y}}, \mathcal{V}_{\text{tst}}) > \eta) = \Pi_{\hat{\boldsymbol{Y}}' \in \hat{\mathcal{Y}}'} \Pr(\pi(\hat{\boldsymbol{Y}}', \mathcal{V}_{\text{tst}}) > \eta) < 0.5^{|\hat{\mathcal{Y}}'|}. \qquad (19)$$

$\square$

### A.9 Rationale of Each Theoretical Result

Here we provide an explanation below about the rationale of each theoretical result in this paper.

**Theorem 1. (Certified Fairness Defense for Node Attributes)**: This theorem gives a way to compute the perturbation-invariant budget (i.e., the budget within which the fairness level will not reduce under a certain threshold) of node attributes. However, since we consider both input data modalities could be attacked, we still need to extend the analysis over the span of node attributes and graph topology (see Theorem 4).

**Theorem 2. (Certified Defense Budget for Structure Perturbations)**: This theorem formulates an optimization problem, whose solution is the perturbation-invariant budget (i.e., the budget within which the fairness level will not reduce under a certain threshold) of graph topology under the smoothed node attributes. However, to solve this optimization problem, we need to explicitly compute $\underline{P_{\tilde{g}_{\boldsymbol{X}}=1}}$ (see Theorem 3).

**Theorem 3. (Positive Probability Lower Bound)**: This theorem provides a way to explicitly compute $\underline{P_{\tilde{g}_{\boldsymbol{X}}=1}}$, which directly enables us to solve the optimization problem in Theorem 2.

**Theorem 4. (Certified Defense Budget for Attribute Perturbations)**: This theorem is built upon Theorem 1 and provides a way to explicitly compute the perturbation-invariant budget of node attributes over the span of node attributes and graph topology.

**Lemma 1. (Perturbation-Invariant Budgets Existence)**: This lemma claims the existence and tractability of the perturbation-invariant budgets on both data modalities, which is further detailed by Theorem 2 and Theorem 4.

**Lemma 2. (Positive Probability Bound Under Noises)**: This lemma claims the existence and tractability of $\underline{P_{\tilde{g}_{\boldsymbol{X}}=1}}$, which is further detailed by Theorem 3.

**Proposition 1. (Probabilistic Guarantee for the Fairness Level of Node Classification)**: This proposition provides a neat probabilistic theoretical guarantee — we have a probability that is large enough to successfully achieve certified defense on fairness.

## B  REPRODUCIBILITY AND SUPPLEMENTARY ANALYSIS

In this section, our primary emphasis is on ensuring the replicability of our experiments, which serves as an extension to Section 4. To begin with, we offer a comprehensive introduction of the three real-world datasets adopted in our experiments. Subsequently, we introduce the detailed experimental settings, as well as the implementation details of our proposed framework, ELEGANT, alongside GNNs and baseline models. Moreover, we outline those essential packages, including their versions, that were utilized in our experiments. Lastly, we elaborate on the supplementary analysis on the time complexity of ELEGANT.

### B.1  DATASETS

In our experiments, we adopt three real-world network datasets that are widely used to perform studies on the fairness of GNNs, namely German Credit (Asuncion & Newman, 2007; Agarwal et al., 2021), Recidivism (Jordan & Freiburger, 2015; Agarwal et al., 2021), and Credit Defaulter (Yeh & Lien, 2009; Agarwal et al., 2021). We introduce their basic information below.

**(1) German Credit.** Each node is a client in a German bank (Asuncion & Newman, 2007), while each edge between any two clients represents that they bear similar credit accounts. Here the gender of bank clients is considered as the sensitive attribute, and the task is to classify the credit risk of the clients as high or low.

**(2) Recidivism.** Each node denotes a defendant released on bail at the U.S state courts during 1990-2009 (Jordan & Freiburger, 2015), and defendants are connected based on the similarity of their past criminal records and demographics. Here the race of defendants is considered as the sensitive attribute, and the task is to classify defendants into more likely vs. less likely to commit a violent crime after being released.

**(3) Credit Defaulter.** This dataset contains credit card users collected from financial agencies (Yeh & Lien, 2009). Specifically, each node in this network denotes a credit card user, and users are connected based on their spending and payment patterns. The sensitive attribute is the age period of users, and the task is to predict the future default of credit card for these users. We present the statistics pf the three datasets above in Table 2.

Table 2: The statistics and basic information about the six real-world datasets adopted for experimental evaluation. Sens. represents the semantic meaning of sensitive attribute.

| Dataset | German Credit | Recidivism | Credit Defaulter |
|---|---|---|---|
| **# Nodes** | 1,000 | 18,876 | 30,000 |
| **# Edges** | 22,242 | 321,308 | 1,436,858 |
| **# Attributes** | 27 | 18 | 13 |
| **Avg. degree** | 44.5 | 34.0 | 95.8 |
| **Sens.** | Gender | Race | Age |
| **Label** | Credit status | Bail decision | Future default |

For the three real-world datasets used in this paper, we adopt the split rate for the training set and validation set as 0.4 and 0.55, respectively. The input node features are normalized before they are fed into the GNNs and the corresponding explanation models. For the downstream task *node classification*, only the labels of the nodes in the training set is available for all models during the training process. The trained GNN models with the best performance on the validation set are preserved for test and explanation.

## B.2 DETAILED EXPERIMENTAL SETTINGS

**Implementation of GNN Models.** In our experiments, all GNN models are implemented in PyTorch (Paszke et al., 2017) with PyG (PyTorch Geometric) (Fey & Lenssen, 2019). For the corresponding hyper-parameters, we set the value of weight decay as 5e-4, with the hidden dimension number and dropout rate being 64 and 0.6, respectively. In addition, we set the learning rate and epoch number as 5e-2 and 200 for training.

**Implementation of ELEGANT.** ELEGANT is implemented in PyTorch (Paszke et al., 2017) with MIT license and all GNNs under ELEGANT are optimized through Adam optimizer (Kingma & Ba, 2015) on Nvidia A6000. In our experiments, the sampling sizes of Gaussian noise and Bernoulli noise are 150 and 200, respectively. All hyper-parameters for GNNs under ELEGANT are set as the same values as the hyper-parameters adopted for vanilla GNNs. We propose to add Gaussian and Bernoulli noise (to node attributes and graph topology) during training, which empirically leads to better certification performance, i.e., larger certification budgets over both node attributes and graph topology. Specifically, we set the entry-wise probability of flipping the existence of an edge and the standard deviation of the added Gaussian noise as 2e-4 and 2e-5, respectively. In addition, we set the confidence level as 0.7 for estimation, since a lower confidence level helps exhibit a clearer tendency of the change of certified budgets w.r.t. other parameters under a limited number of sampling size, considering the computational costs. In the test phase, we set the sampled ratio for certification (from the nodes out of training and validation set) to be 0.9 to make the sampled size relatively large, in which way we include more nodes in the set of nodes to be certified. In each run, we sample 100 times, and the value of FCR is averaged across three runs with different seeds. Finally, considering the sizes of the three datasets, we set the nodes that are vulnerable to be 5% for German Credit and 1% for others.

**Selection of $\epsilon$ and $\beta$.** There are two critical parameters, $\epsilon$ and $\beta$, that could affect the effectiveness of ELEGANT. These two parameters control the level of randomness for the added Gaussian and Bernoulli noise, respectively. Intuitively, larger $\epsilon$ and $\beta$ will induce more randomness in the node

attributes and graph structure, which could make ELEGANT more robust to perturbations with larger sizes and thus achieve larger $\epsilon_X$ and $\epsilon_A$. However, if $\epsilon_X$ and $\epsilon_A$ are too large, the randomness could go beyond what the GNN classifier can manage and could finally cause failure in certification. Hence it is necessary to first determine appropriate values of $\epsilon_X$ and $\epsilon_A$ for ELEGANT. Here we propose a strategy for parameter selection to realize as large certified defense budgets as possible. Specifically, we first set an empirical $\eta$ to be 25% higher than the fairness level of the corresponding vanilla GNN model. Such a threshold calibrates across different GNNs and can be considered as a reasonable threshold for the exhibited bias. Then we determine two wide search spaces for $\sigma$ and $\beta$, respectively, and compute the averaged $\epsilon_X$ and $\epsilon_A$ from multiple runs over each pair of $\sigma$ and $\beta$ values. We now rank $(\sigma, \beta)$ pairs based on the averaged $\epsilon_X$ and $\epsilon_A$ in a descending order, respectively. Finally, we truncate the obtained two rankings from their most top-ranked $(\sigma, \beta)$ pair to the tail, until the two truncated rankings have the first overlapped $(\sigma, \beta)$ pair. Such an identified $(\sigma, \beta)$ pair can achieve large and balanced certification budgets over both $A$ and $X$, and hence they are recommended.

**Implementation of Baseline Models.** In this paper, we include two fairness-aware GNNs as the baselines for comparison, namely FairGNN and NIFTY. We introduce the details below. (1) **FairGNN.** For FairGNN, we adopt the official implementations from (Dai & Wang, 2021). Hyper-parameters corresponding to the GNN model structure (such as the number of hidden dimensions) are ensured to be the same as the vanilla GNNs for a fair comparison. Other parameters are carefully tuned under the guidance of the recommended training settings. (2) **NIFTY.** For NIFTY, we use the official implementations provided from (Agarwal et al., 2021). We ensured that the parameters related to the GNN model structure stay the same as the original GNNs for a fair comparison. We also adjust other parameters based on the suggested training settings for better performance.

**Packages Required for Implementations.** We list those key packages and their corresponding versions adopted in our experiments below.

- Python == 3.8.8

- torch == 1.10.1

- torch-geometric == 2.1.0

- torch-scatter == 2.0.9

- torch-sparse == 0.6.13

- cuda == 11.1

- numpy == 1.20.1

- tensorboard == 2.10.0

- networkx == 2.5

- scikit-learn == 0.24.2

- pandas==1.2.4

- scipy==1.6.2

### B.3 ALGORITHMIC ROUTINE

Now we introduce the pipeline of the proposed framework ELEGANT to obtain the node classification results in facing of the graph data that could have been perturbed by malicious attackers. We present the algorithmic routine in Algorithm 1. Note that ABSTAIN refers to the case where certification fails. Correspondingly, FCR measures the ratio of not returning ABSTAIN for the proposed framework ELEGANT, which generally reflects the usability of the certification defense.

## B.4    EVALUATION OF MODEL UTILITY

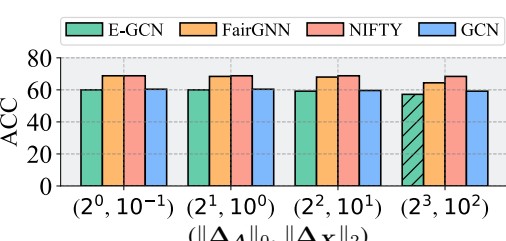

Figure 4: The utility of GCN, E-GCN, FairGNN, and NIFTY under fairness attacks on German Credit. The shaded bar indicates that certified budget $\epsilon_A \leq \|\mathbf{\Delta}_A\|_0$ or $\epsilon_X \leq \|\mathbf{\Delta}_X\|_2$.

In Section 4.3, we present the comparison between ELEGANT and baseline models over the fairness level under attacks. We now present the comparison over the utility under attacks. Specifically, we utilize node classification accuracy as the indicator of model utility, and we present the results in Figure 4. The fairness-aware GNNs are found to exhibit better utility compared with the vanilla GNNs, which is a common observation consistent with a series of existing works (Agarwal et al., 2021; Dong et al., 2022a). More importantly, we observe that the ELEGANT does not jeopardize the performance of GNN compared with the utility of the vanilla GNN. This demonstrate a high level of usability for ELEGANT in real-world applications.

## B.5    CERTIFICATION UNDER DIFFERENT FAIRNESS METRICS

In Section 4.2, we present the experimental results based on the fairness metric of $\Delta_{\text{SP}}$, which measures the exhibited bias under the fairness notion of *Statistical Parity*. We also perform the experiments based on $\Delta_{\text{EO}}$, which measures the exhibited bias under the fairness notion of *Equal Opportunity*. We present the experimental results in Table 3. We summarize the observations below. (1) **Fairness Certification Rate (FCR).** We observe that ELEGANT realizes large values of FCR (larger than 80%) for all three GNN backbones and three attributed network datasets. Similar to our discussion in Section 4.2, this demonstrate that for nodes in any randomly sampled test set, we have a probability around or larger than 80% to successfully certify the fairness level of the predictions yielded by the GNN model with our proposed framework ELEGANT. As a consequence, we argue that ELEGANT also achieves a satisfying fairness certification rate across all adopted GNN backbones and datasets on the basis of $\Delta_{\text{EO}}$. In addition, we also observe that the German Credit dataset bears relatively larger values of FCR, while the values of FCR are relatively smaller with relatively larger standard deviation values on Recidivism and Credit Defaulter datasets. A possible reason is that we set the threshold (i.e., $\eta$) as a value 25% higher than the bias exhibited by the vanilla GNNs. Consequently, if the vanilla GNNs already exhibit a low level of bias, the threshold determined with such a strategy could be hard to satisfy under the added noise. This evidence indicates that the proposed framework ELEGANT tends to deliver better performance under scenarios where vanilla GNNs exhibit a high level of bias with the proposed strategy. (2) **Utility.** Compared with vanilla GNNs, certified GNNs with ELEGANT exhibit comparable and even higher node classification accuracy values in all cases. Therefore, we argue that the proposed framework ELEGANT does not significantly jeopardize the utility of the vanilla GNN models in certifying the fairness level of node classification. (3) **Fairness.** We observe that certified GNNs with ELEGANT are able to achieve better performances in terms of algorithmic fairness compared with those vanilla GNNs. This evidence indicates that the proposed framework ELEGANT also helps to mitigate the exhibited bias (by the backbone GNN models). We conjecture that such bias mitigation should be attributed to the same reason discussed in Section 4.2.

Table 3: Comparison between vanilla GNNs and certified GNNs under ELEGANT over three popular GNNs across three real-world datasets. Here ACC is node classification accuracy, and E- prefix marks out the GNNs under ELEGANT with certification. ↑ denotes the larger, the better; ↓ denotes the opposite. Different from the table in Section 4.2 (where the bias is measured with $\Delta_{\text{SP}}$), the bias is measured with $\Delta_{\text{EO}}$ here. Numerical values are in percentage, and the best ones are in bold.

| | German Credit | | | Recidivism | | | Credit Defaulter | | |
|---|---|---|---|---|---|---|---|---|---|
| | **ACC (↑)** | **Bias (↓)** | **FCR (↑)** | **ACC (↑)** | **Bias (↓)** | **FCR (↑)** | **ACC (↑)** | **Bias (↓)** | **FCR (↑)** |
| **SAGE** | $67.3_{\pm 2.14}$ | $41.8_{\pm 11.0}$ | N/A | $89.8_{\pm 0.66}$ | $6.09_{\pm 3.10}$ | N/A | $75.9_{\pm 2.18}$ | $10.4_{\pm 1.59}$ | N/A |
| **E-SAGE** | $\mathbf{72.2}_{\pm 1.26}$ | $\mathbf{8.63}_{\pm 6.15}$ | $100_{\pm 0.00}$ | $\mathbf{90.8}_{\pm 0.97}$ | $3.12_{\pm 3.64}$ | $81.0_{\pm 13.0}$ | $73.4_{\pm 0.61}$ | $\mathbf{7.18}_{\pm 1.06}$ | $88.7_{\pm 6.02}$ |
| **GCN** | $\mathbf{59.6}_{\pm 3.64}$ | $35.0_{\pm 4.77}$ | N/A | $90.5_{\pm 0.73}$ | $6.35_{\pm 1.65}$ | N/A | $\mathbf{65.8}_{\pm 0.29}$ | $13.5_{\pm 4.23}$ | N/A |
| **E-GCN** | $58.8_{\pm 3.74}$ | $\mathbf{29.8}_{\pm 6.82}$ | $93.3_{\pm 8.73}$ | $89.3_{\pm 0.92}$ | $\mathbf{3.93}_{\pm 3.12}$ | $96.0_{\pm 4.97}$ | $63.5_{\pm 0.37}$ | $\mathbf{9.12}_{\pm 0.95}$ | $80.5_{\pm 14.5}$ |
| **JK** | $63.3_{\pm 4.11}$ | $37.7_{\pm 15.9}$ | N/A | $\mathbf{91.9}_{\pm 0.54}$ | $5.26_{\pm 3.25}$ | N/A | $76.6_{\pm 0.69}$ | $8.04_{\pm 0.57}$ | N/A |
| **E-JK** | $\mathbf{63.4}_{\pm 3.68}$ | $\mathbf{31.2}_{\pm 15.5}$ | $93.7_{\pm 8.96}$ | $90.1_{\pm 0.55}$ | $\mathbf{2.54}_{\pm 1.62}$ | $83.7_{\pm 8.96}$ | $\mathbf{76.9}_{\pm 0.86}$ | $\mathbf{2.90}_{\pm 2.04}$ | $95.7_{\pm 4.80}$ |

---

**Algorithm 1** Certified Defense on the Fairness of GNNs

---

**Input:**
   $\mathcal{G}$: graph data with potential malicious attacks; $f_{\boldsymbol{\theta}*}$: an optimized GNN node classifier; $\mathcal{V}_{\text{train}}$, $\mathcal{V}_{\text{validation}}$, $\mathcal{V}_{\text{test}} \in \mathcal{V}$: the node set for training, validation, and test, respectively; $\mathcal{V}_{\text{vul}} \in \mathcal{V}_{\text{test}}$: the set of vulnerable nodes that may bear attacks (on node attributes and/or graph topology); $N_1$, $N_2$: sample size for the set of Bernoulli and Gaussian noise, respectively; $\eta$: a given threshold for the exhibited bias; $\alpha$: the parameter to indicate the confidence level $(1 - \alpha)$ of the estimation; $\sigma$: the std of the added Gaussian noise; $\beta$: the probability of returning zero of the added Bernoulli noise;

**Output:**
   $\epsilon_{\boldsymbol{A}}$: the certified defense budget over the adjacency matrix $\boldsymbol{A}$; $\epsilon_{\boldsymbol{X}}$: the certified defense budget over the node attribute matrix $\boldsymbol{X}$; $\hat{\boldsymbol{Y}}'$: the output node classification results from the certified classifier;

 1: Sample a set of Bernoulli noise $\mathcal{Q}_{\text{B}}$ containing $N_1$ samples;
 2: Sample a set of Gaussian noise $\mathcal{Q}_{\text{G}}$ containing $N_2$ samples;
 3: **for** $\omega_{\boldsymbol{A}} \in \mathcal{Q}_{\text{B}}$ **do**
 4:     **for** $\omega_{\boldsymbol{X}} \in \mathcal{Q}_{\text{G}}$ **do**
 5:         Calculate and collect the output of $f_{\boldsymbol{\theta}*}$ under the noise of $\omega_{\boldsymbol{A}}$ and $\omega_{\boldsymbol{X}}$;
 6:         Calculate and collect the output of $g$ based on the output of $f_{\boldsymbol{\theta}*}$;
 7:     **end for**
 8:     Under $\mathcal{Q}_{\text{G}}$, collect the number of $g$ returning 1 and 0 as $n_1$ and $n_0$, respectively;
 9:     Estimate the lower bound of returning $c$ as $\underline{P_{g=c}}$ determined by the larger one between $n_1$ and $n_0$;
10:     **if** $n_1 > n_0$ and $\underline{P_{g=1}}$ is larger than 0.5 with a confidence level larger than $1 - \alpha$ **or** $n_1 < n_0$ and $\underline{P_{g=0}}$ is larger than 0.5 with a confidence level larger than $1 - \alpha$ **then**
11:         Calculate and collect the value of $\tilde{\epsilon}_{\boldsymbol{X}}$;
12:     **else**
13:         **return** ABSTAIN
14:     **end if**
15: **end for**
16: Collect the number of cases where $n_1 > n_0$ and estimate the lower bound of returning 1 as $\underline{P_{\tilde{g}_{\boldsymbol{X}}=1}}$;
17: **if** $\underline{P_{\tilde{g}_{\boldsymbol{X}}=1}}$ is larger than 0.5 with a confidence level larger than $1 - \alpha$ **then**
18:     Calculate $\epsilon_{\boldsymbol{X}}$ (out of the collected $\tilde{\epsilon}_{\boldsymbol{X}}$) and $\epsilon_{\boldsymbol{A}}$ (based on the estimated $\underline{P_{\tilde{g}_{\boldsymbol{X}}=1}}$);
19:     Find $\boldsymbol{Y}'$ out of the collected output of $f_{\boldsymbol{\theta}*}$;
20:     **return** $\boldsymbol{Y}'$, $\epsilon_{\boldsymbol{X}}$, and $\epsilon_{\boldsymbol{A}}$;
21: **else**
22:     **return** ABSTAIN
23: **end if**

---

### B.6 ORDERING THE INNER AND OUTER DEFENSE

We first review the general pipeline to achieve certified fairness defense. Specifically, we first model the fairness attack and defense by formulating the bias indicator function $g$. Then, we achieve certified defense over the node attributes for $g$, which leads to classifier $\tilde{g}_{\boldsymbol{X}}$. Finally, we realize certified defense for $\tilde{g}_{\boldsymbol{X}}$ over the graph topology, which leads to classifier $\tilde{g}_{\boldsymbol{A},\boldsymbol{X}}$. In general, we may consider the certified defense over node attributes and graph topology as the inner certified classifier and outer certified classifier, respectively. Now, a natural question is: *is it possible to achieve certified defense in a different order, i.e., first achieve certified defense over the graph topology (as the inner classifier), and then realize certified defense over the node attributes (as the outer classifier)?* Note that this is not the research focus of this paper, but we will provide insights about this question. In fact, it is also feasible to achieve certified defense in the reversed order compared with the approach presented in our paper. We provide an illustration in Figure 5. We follow a similar setting to plot this figure as in Section 3.3. Specifically, in case (1), both $\boldsymbol{A}_{i,j} \oplus 0$ and $\boldsymbol{A}_{i,j} \oplus 1$ lead to a positive outcome for $g$; in case (2), both $\boldsymbol{A}_{i,j} \oplus 0$ and $\boldsymbol{A}_{i,j} \oplus 1$ lead to a negative outcome. However, considering the Gaussian distribution around $\boldsymbol{X}_{i,j}$, samples will fall around case (1) with a much higher number compared with case (2).

Hence, in this example, it would be reasonable to assume that the classifier with Bernoulli noise over graph topology (the inner certified classifier) will return 1 with a higher probability. This example thus illustrates how certification following a different order returns 1.

However, such a formulation bears higher computational costs in calculating the certified budgets. The reason is that we are able to utilize a closed-form solution to calculate $\epsilon_{\boldsymbol{X}}$ based on a set of Gaussian noise and the corresponding output from the bias indicator function. However, based on a set of Bernoulli noise and the corresponding output from the bias indicator function, we will need to solve the optimization problem given in Theorem 2 to calculate $\epsilon_{\boldsymbol{A}}$, which bears a higher time complexity than calculating $\epsilon_{\boldsymbol{X}}$. If we follow the strategy provided in Section 3.4 to calculate the inner and outer certification budgets, the certified budget of the inner certification will always

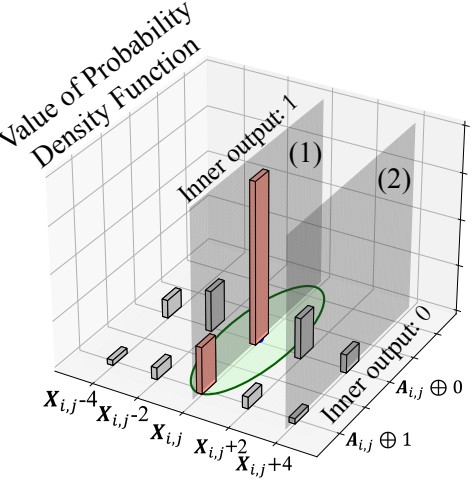

Figure 5: An example illustrating how ELE-GANT works with a different order to achieve certified defense.

be calculated multiple times, while the certified budget of the outer certification will only be calculated once. Considering the high computational cost of calculating $\epsilon_{\boldsymbol{A}}$, we thus argue that it is more efficient to realize the certification over graph topology as the outer certified classifier.

### B.7 CERTIFICATION WITH ESTIMATED PROBABILITIES

In Section 3.4, we proposed to utilize estimated lower bounds of the probabilities (including $P(c)$ in Theorem 1 and $\Pr(\tilde{g}_{\boldsymbol{X}}(\boldsymbol{A} \oplus \boldsymbol{\Gamma}_{\boldsymbol{A}}, \boldsymbol{X}) = 1)$ in Theorem 3) to perform certification in practice, considering the exact probability values are difficult to compute. In Appendix A.1 and Appendix A.5, we have discussed that both theorems hold no matter exact probability values or estimated lower bounds (of the probabilities above) are used. Now we present a brief review of other theoretical analysis to show that they also hold. (1) for Lemma 2, note that taking a lower bound estimation to replace the exact $\Pr(\tilde{g}_{\boldsymbol{X}}(\boldsymbol{A} \oplus \boldsymbol{\Gamma}_{\boldsymbol{A}}, \boldsymbol{X}) = 1)$ reduces the total size of $\mathcal{H}$ in Theorem 3. Correspondingly, the formulated $\underline{P_{\tilde{g}_{\boldsymbol{X}}=1}}$ based on the estimated $\Pr(\tilde{g}_{\boldsymbol{X}}(\boldsymbol{A} \oplus \boldsymbol{\Gamma}_{\boldsymbol{A}}, \boldsymbol{X}) = 1)$ is smaller than that based on

the exact $\Pr(\tilde{g}_{\boldsymbol{X}}(\boldsymbol{A} \oplus \boldsymbol{\Gamma}_{\boldsymbol{A}}, \boldsymbol{X}) = 1)$. Hence Lemma 2 still holds when $P_{\tilde{g}_{\boldsymbol{X}}=1}$ is replaced with one calculated based on the estimated $\Pr(\tilde{g}_{\boldsymbol{X}}(\boldsymbol{A} \oplus \boldsymbol{\Gamma}_{\boldsymbol{A}}, \boldsymbol{X}) = 1)$. (2) For Theorem 2, it holds no matter how $P(c)$ in Theorem 1 and $\Pr(\tilde{g}_{\boldsymbol{X}}(\boldsymbol{A} \oplus \boldsymbol{\Gamma}_{\boldsymbol{A}}, \boldsymbol{X}) = 1)$ in Theorem 3 are obtained. (3) For Theorem 4, according to Appendix A.7, it still holds as long as Theorem 1 holds. (4) For Proposition 1, in all cases where $\Pr(g(\boldsymbol{A} \oplus \boldsymbol{\Gamma}'_{\boldsymbol{A}}, \boldsymbol{X} + \boldsymbol{\Gamma}_{\boldsymbol{X}}) = 1)$ is identified to be larger than 0.5 with an estimated lower bound, the (underlying) exact $\Pr(g(\boldsymbol{A} \oplus \boldsymbol{\Gamma}'_{\boldsymbol{A}}, \boldsymbol{X} + \boldsymbol{\Gamma}_{\boldsymbol{X}}) = 1)$ will be larger than the estimated probability value under the given confidence level, and thus will also be larger than 0.5. Here $\boldsymbol{\Gamma}'_{\boldsymbol{A}}$ is a sampled Bernoulli noise, i.e., $\boldsymbol{\Gamma}'_{\boldsymbol{A}} \in \bar{\mathcal{A}}'$. According to Appendix A.8, Proposition 1 still holds in this case. (5) Finally, we conclude that Lemma 1 still holds since Theorem 1, Lemma 2, Theorem 3, and Theorem 4 hold.

### B.8    TIME COMPLEXITY ANALYSIS

We now present a comprehensive analysis on the time complexity of ELEGANT. We present the analysis from both theoretical and experimental perspectives.

**Theoretical.** The time complexity is linear w.r.t. the total number of the random perturbations $N$, i.e., $\mathcal{O}(N)$. We perform 30,000 random perturbations over the span of node attributes and graph structure. We note that the actual running time is acceptable since the certification does not require re-training (which is the most costly process). In addition, all runnings do not rely on the prediction results from each other. Hence they can be paralleled altogether theoretically to further reduce the running time.

**Experimental.** We perform a study of running time, and we present the results in Table 4. Specifically, we compare the running time of a successful certification under 30,000 random noise samples and a regular training-inference cycle with vanilla GCN. We observe that (1) although ELEGANT improves the computational cost compared with the vanilla GNN backbones, the running time remains acceptable; and (2) ELEGANT has less running time growth rate on larger datasets. For example, E-SAGE has around 10x running time on German Credit (a smaller dataset) while only around 4x on Credit Default (a larger dataset) compared to vanilla SAGE. Hence we argue that ELEGANT bears a high level of usability in terms of complexity and running time.

### B.9    ADDITIONAL RESULTS ON DIFFERENT GNN BACKBONES & BASELINES

We perform additional experiments over two popular GNNs, including APPNP (Klicpera et al., 2019) and GCNII (Chen et al., 2020), to evaluate the generalization ability of ELEGANT onto different backbones. We present all numerical results in Table 5 (in terms of accuracy), Table 6 (in terms of fairness), and Table 7 (in terms of FCR). We observe that ELEGANT achieves comparable utility, a superior level of fairness, and a large percentage of FCR. This verifies the satisfying usability of ELEGANT, which remains consistent with the paper.

In addition, we provide a detailed fairness comparison between ELEGANT and robust GNNs from (Jin et al., 2021) and (Wu et al., 2019b) in Table 8. We observe that the best performances still come from the GNNs equipped with ELEGANT on all datasets. Hence we argue that ELEGANT exhibits satisfying performance in usability, which remains consistent with the discussion in the paper.

**Why ELEGANT Improves Fairness?** We note that improving fairness is a byproduct of ELEGANT, and our focus is to achieve certification over the fairness level of the prediction results. We now provide a detailed discussion about why fairness is improved here. First, existing works found that the distribution difference in the node attribute values and edge existence across different subgroups is a significant source of bias (Dong et al., 2022a; Dai & Wang, 2021; Fan et al., 2021). However, adding noise on both node attributes and graph topology may reduce such distributional divergence and mitigate bias. Second, As mentioned in Section 3.4, the proposed strategy to obtain the output

predictions in ELEGANT is to select the fairest result among the output set $\hat{\mathcal{Y}}'$, where each output is derived based on a sample $\mathbf{\Gamma}'_A \in \bar{\mathcal{A}}'$ (i.e., $\mathrm{argmin}_{\hat{Y}'}, \pi(\hat{Y}', \mathcal{V}_{\mathrm{tst}})$ $s.t.$ $\hat{Y}' \in \hat{\mathcal{Y}}'$). Such a strategy provides a large enough probability to achieve certification in light of Proposition 1. Meanwhile, we point out that such a strategy also helps to significantly improve fairness since highly biased outputs are excluded.

### B.10 Complementary Results

We provide the results in terms of $\Delta_{EO}$ for Table 1 in Table 9, and we present the results of the baselines for Figure 2 in Table 10, Table 11, Table 12, and Table 13. For Table 9, we observe that ELEGANT does not constantly show a lower value of $\Delta_{EO}$. This is because the certification goal in Table 1 is $\Delta_{SP}$ instead of $\Delta_{EO}$. In addition, we note that debiasing existing GNN models is not the goal of this paper. In addition, we provide the corresponding results in terms of accuracy for Figure 3 in Table 14 and Table 15. We observe that although most performance remains stable, a stronger noise (i.e., larger $\sigma$ and smaller $\beta$) generally leads to worse but still comparable performance. This is consistent with the discussion in Section 4.4, and this has been taken into consideration in the discussion of the parameter selection strategy in Section 4.4.

Table 4: Comparison of running time (in seconds) on different datasets using different methods.

|  | German | Recidivism | Credit |
| --- | --- | --- | --- |
| **SAGE** | $5.27 \pm 0.38$ | $34.14 \pm 1.08$ | $40.11 \pm 0.36$ |
| **E-SAGE** | $\mathbf{53.23 \pm 1.31}$ | $\mathbf{137.12 \pm 58.66}$ | $\mathbf{157.51 \pm 37.21}$ |
| **GCN** | $5.59 \pm 0.37$ | $34.94 \pm 1.16$ | $40.59 \pm 0.32$ |
| **E-GCN** | $\mathbf{53.79 \pm 30.19}$ | $\mathbf{212.94 \pm 10.38}$ | $\mathbf{214.11 \pm 10.31}$ |
| **JK** | $5.78 \pm 0.43$ | $34.68 \pm 0.88$ | $39.44 \pm 1.56$ |
| **E-JK** | $\mathbf{59.99 \pm 25.01}$ | $\mathbf{238.37 \pm 1.81}$ | $\mathbf{252.99 \pm 17.03}$ |

Table 5: Performance comparison of classification accuracy. Numbers are in percentage.

|  | German | Recidivism | Credit |
| --- | --- | --- | --- |
| **SAGE** | $67.3 \pm 2.14$ | $89.8 \pm 0.66$ | $75.9 \pm 2.18$ |
| **E-SAGE** | $\mathbf{71.0 \pm 1.27}$ | $\mathbf{89.9 \pm 0.90}$ | $\mathbf{73.4 \pm 0.50}$ |
| **GCN** | $59.6 \pm 3.64$ | $90.5 \pm 0.73$ | $65.8 \pm 0.29$ |
| **E-GCN** | $\mathbf{58.2 \pm 1.82}$ | $\mathbf{89.6 \pm 0.74}$ | $\mathbf{65.2 \pm 0.99}$ |
| **JK** | $63.3 \pm 4.11$ | $91.9 \pm 0.54$ | $76.6 \pm 0.69$ |
| **E-JK** | $\mathbf{62.3 \pm 4.07}$ | $\mathbf{89.3 \pm 0.33}$ | $\mathbf{77.7 \pm 0.27}$ |
| **APPNP** | $69.9 \pm 2.17$ | $95.3 \pm 0.78$ | $74.4 \pm 3.05$ |
| **E-APPNP** | $\mathbf{69.4 \pm 0.83}$ | $\mathbf{95.9 \pm 0.02}$ | $\mathbf{74.6 \pm 0.32}$ |
| **GCNII** | $60.9 \pm 1.00$ | $90.4 \pm 0.95$ | $77.7 \pm 0.22$ |
| **E-GCNII** | $\mathbf{60.4 \pm 4.45}$ | $\mathbf{88.8 \pm 0.24}$ | $\mathbf{77.6 \pm 0.02}$ |

## C Additional Discussion

### C.1 Why Certify A Classifier on top of An Optimized GNN?

We note that the rationale of certified defense is to provably maintain the classification results against attacks. Under this context, most existing works on certifying an existing deep learning model focus

Table 6: Comparison of fairness (measured with $\Delta_{SP}$). Numbers are in percentage.

|  | German | Recidivism | Credit |
|---|---|---|---|
| SAGE | $50.6 \pm 15.9$ | $9.36 \pm 3.15$ | $13.0 \pm 4.01$ |
| E-SAGE | $\mathbf{16.3 \pm 10.9}$ | $\mathbf{6.39 \pm 2.85}$ | $\mathbf{8.94 \pm 0.99}$ |
| GCN | $37.4 \pm 3.24$ | $10.1 \pm 3.01$ | $11.1 \pm 3.22$ |
| E-GCN | $\mathbf{3.52 \pm 3.77}$ | $\mathbf{9.56 \pm 3.22}$ | $\mathbf{7.28 \pm 1.46}$ |
| JK | $41.2 \pm 18.1$ | $10.1 \pm 3.15$ | $9.24 \pm 0.60$ |
| E-JK | $\mathbf{22.4 \pm 1.95}$ | $\mathbf{6.26 \pm 2.78}$ | $\mathbf{3.37 \pm 2.64}$ |
| APPNP | $27.4 \pm 4.81$ | $9.71 \pm 3.57$ | $12.3 \pm 3.14$ |
| E-APPNP | $\mathbf{13.1 \pm 5.97}$ | $\mathbf{2.23 \pm 0.04}$ | $\mathbf{10.8 \pm 0.07}$ |
| GCNII | $51.4 \pm 0.36$ | $9.70 \pm 3.37$ | $7.62 \pm 0.29$ |
| E-GCNII | $\mathbf{24.9 \pm 0.47}$ | $\mathbf{3.78 \pm 0.93}$ | $\mathbf{1.72 \pm 0.81}$ |

Table 7: Performance in FCR on different datasets and backbone GNNs. Numbers are in percentage.

|  | German | Recidivism | Credit |
|---|---|---|---|
| E-SAGE | $98.7 \pm 1.89$ | $94.3 \pm 6.65$ | $94.3 \pm 3.3$ |
| E-GCN | $96.3 \pm 1.89$ | $96.0 \pm 3.56$ | $92.7 \pm 5.19$ |
| E-JK | $97.0 \pm 3.00$ | $89.5 \pm 10.5$ | $99.3 \pm 0.47$ |
| E-APPNP | $97.8 \pm 3.14$ | $87.1 \pm 3.79$ | $95.5 \pm 6.43$ |
| E-GCNII | $94.7 \pm 5.27$ | $92.9 \pm 9.93$ | $99.0 \pm 1.41$ |

on certifying a specific predicted label over a given data point. Here, the prediction results to be certified are classification results. Correspondingly, these works are able to certify the model itself.

However, the strategy above is not feasible in our studied problem. This is because we seek to certify the level of fairness of a group of nodes. The value of such a group-level property cannot be directly considered as a classification result, and thus they are not feasible to be directly certified. Therefore, we proposed to first formulate a classifier on top of an optimized GNN. As such, achieving certification becomes feasible. In fact, this also serves as one of the contributions of our work.

## C.2 WHAT IS THE DIFFERENCE BETWEEN THE ATTACKING PERFORMANCE OF GNNS AND THE FAIRNESS OF GNNS?

In traditional attacks over the performance of GNNs, the objective of the attacker is simply formulated as having false predictions on as many nodes as possible, such that the overall performance is jeopardized. However, in attacks over the fairness of GNNs, whether the goal of the attacker can be achieved is jointly determined by the GNN predictions over all nodes. Such node-level dependency in achieving the attacking goal makes the defense over fairness attacks more difficult, since the defense cannot be directly performed at the node level but at the model level instead. Correspondingly, this necessitates (1) constructing an additional classifier as discussed in the previous reply, and (2) additional theoretical analysis over the constructed classifier as in Theorem 1, 2, and 3 to achieve certification.

Table 8: Comparison of fairness (measured with $\Delta_{SP}$). Numbers are in percentage.

|  | German | Recidivism | Credit |
|---|---|---|---|
| **SAGE** | $50.6 \pm 15.9$ | $9.36 \pm 3.15$ | $13.0 \pm 4.01$ |
| **E-SAGE** | $\mathbf{16.3 \pm 10.9}$ | $\mathbf{6.39 \pm 2.85}$ | $\mathbf{8.94 \pm 0.99}$ |
| **GCN** | $37.4 \pm 3.24$ | $10.1 \pm 3.01$ | $11.1 \pm 3.22$ |
| **E-GCN** | $\mathbf{3.52 \pm 3.77}$ | $\mathbf{9.56 \pm 3.22}$ | $\mathbf{7.28 \pm 1.46}$ |
| **JK** | $41.2 \pm 18.1$ | $10.1 \pm 3.15$ | $9.24 \pm 0.60$ |
| **E-JK** | $\mathbf{22.4 \pm 1.95}$ | $\mathbf{6.26 \pm 2.78}$ | $\mathbf{3.37 \pm 2.64}$ |
| (Jin et al., 2021) | $14.8 \pm 18.3$ | $9.59 \pm 0.65$ | $3.84 \pm 0.17$ |
| (Wu et al., 2019b) | $3.66 \pm 0.52$ | $8.04 \pm 2.97$ | $7.10 \pm 5.10$ |

Table 9: The $\Delta_{EO}$ of Table 1 in the paper. All numerical numbers are in percentage.

|  | German | Recidivism | Credit |
|---|---|---|---|
| **SAGE** | $30.43 \pm 0.07$ | $3.71 \pm 0.01$ | $5.56 \pm 0.03$ |
| **E-SAGE** | $\mathbf{12.21 \pm 0.04}$ | $\mathbf{6.95 \pm 0.02}$ | $\mathbf{7.18 \pm 0.01}$ |
| **GCN** | $35.19 \pm 0.07$ | $5.06 \pm 0.01$ | $11.9 \pm 0.02$ |
| **E-GCN** | $\mathbf{8.32 \pm 0.03}$ | $\mathbf{1.39 \pm 0.01}$ | $\mathbf{6.24 \pm 0.02}$ |
| **JK** | $18.10 \pm 0.13$ | $3.02 \pm 0.01$ | $9.47 \pm 0.02$ |
| **E-JK** | $\mathbf{23.68 \pm 0.02}$ | $\mathbf{2.74 \pm 0.01}$ | $\mathbf{2.55 \pm 0.01}$ |

## C.3 CERTIFICATION WITHOUT CONSIDERING THE BINARY SENSITIVE ATTRIBUTE

We utilize the most widely studied setting to assume the sensitive attributes are binary. However, our certification approach is not designed to be tailored to the sensitive attributes. Therefore, our approach can be easily extended to scenarios where the sensitive attributes are multi-class and continuous by adopting the corresponding fairness metric as the function $\pi(\cdot)$ in Definition 1.

## C.4 HOW DO THE MAIN THEORETICAL FINDINGS DIFFER FROM EXISTING WORKS ON ROBUSTNESS CERTIFICATION OF GNNS ON REGULAR ATTACKS?

Most existing works for robustness certification can only defend against attacks on either node attributes or graph structure. Due to the multi-modal input data of GNNs, existing works usually fail to handle the attacks over node attributes and graph structure at the same time. However, ELEGANT is able to defend against attacks over both data modalities. This necessitates using both continuous and discrete noises for smoothing and the analysis for joint certification in the span of the two input data modalities (as shown in Figure 1).

Table 10: The results under $(2^0, 10^{-1})$ in terms of node classification accuracy, AUC score, F1 score, $\Delta_{SP}$, and $\Delta_{EO}$ Figure 2. All numerical numbers are in percentage.

| $(2^0, 10^{-1})$ | **Accuracy** | **AUC** | **F1 Score** | $\Delta_{SP}$ | $\Delta_{EO}$ |
|---|---|---|---|---|---|
| **GCN** | 58.4% | 66.4% | 63.9% | 41.4% | 33.4% |
| **NIFTY** | 61.2% | 68.1% | 66.2% | 33.9% | 13.3% |
| **FairGNN** | 55.2% | 62.2% | 61.4% | 16.4% | 5.99% |

Table 11: The results under $(2^1, 10^0)$ in terms of node classification accuracy, AUC score, F1 score, $\Delta_{SP}$, and $\Delta_{EO}$ Figure 2. All numerical numbers are in percentage.

| $(2^1, 10^0)$ | Accuracy | AUC | F1 Score | $\Delta_{SP}$ | $\Delta_{EO}$ |
|---|---|---|---|---|---|
| GCN | 58.4% | 66.4% | 63.9% | 41.4% | 33.4% |
| NIFTY | 61.2% | 68.2% | 66.2% | 36.1% | 13.3% |
| FairGNN | 55.2% | 62.2% | 61.4% | 16.8% | 7.77% |

Table 12: The results under $(2^2, 10^1)$ in terms of node classification accuracy, AUC score, F1 score, $\Delta_{SP}$, and $\Delta_{EO}$ for Figure 2. All numerical numbers are in percentage.

| $(2^2, 10^1)$ | Accuracy | AUC | F1 Score | $\Delta_{SP}$ | $\Delta_{EO}$ |
|---|---|---|---|---|---|
| GCN | 58.0% | 66.6% | 63.7% | 41.4% | 37.8% |
| NIFTY | 61.2% | 68.1% | 66.0% | 42.1% | 13.3% |
| FairGNN | 55.6% | 62.1% | 61.9% | 16.0% | 9.56% |

## C.5 DISCUSSION: DIFFERENCE WITH EXISTING SIMILAR WORKS

Here we mainly focus on discussing the difference between this work and (Bojchevski et al., 2020).

We note that (1) the randomized smoothing technique adopted in (Bojchevski et al., 2020) is different from the proposed randomized smoothing approach on the graph topology in this paper and (2) the techniques in (Bojchevski et al., 2020) tackle a different problem from this paper. We elaborate on more details below.

The techniques in (Bojchevski et al., 2020) are different from this paper. Although both randomized smoothing approaches are able to handle binary data, we note that the randomized smoothing approach proposed in (Bojchevski et al., 2020) is data-dependent. However, the proposed randomized smoothing approach in this paper is data-independent. We note that in practice, a data-independent approach enables practitioners to pre-generate noises, which significantly improves usability.

The studied problem in (Bojchevski et al., 2020) is different from this paper. Although the authors claimed to achieve a joint certificate for graph topology and node attributes in (Bojchevski et al., 2020), all node attributes are assumed to be binary, which can only be applied to cases where these attributes are constructed as bag-of-words representations (as mentioned in the second last paragraph in the Introduction of (Bojchevski et al., 2020)). However, in this work, we follow a more realistic setting where only graph topology is assumed to be binary while node attributes are considered as continuous. This makes the problem more difficult to handle, since different strategies should be adopted for different data modalities. In summary, compared with (Bojchevski et al., 2020), the problem studied in this paper is more realistic and more suitable for GNNs.

Table 13: The results under $(2^3, 10^2)$ in terms of node classification accuracy, AUC score, F1 score, $\Delta_{SP}$, and $\Delta_{EO}$ for Figure 2. All numerical numbers are in percentage.

| $(2^3, 10^2)$ | Accuracy | AUC | F1 Score | $\Delta_{SP}$ | $\Delta_{EO}$ |
|---|---|---|---|---|---|
| GCN | 58.0% | 67.7% | 63.7% | 42.6% | 45.7% |
| NIFTY | 58.8% | 67.3% | 63.1% | 44.5% | 19.4% |
| FairGNN | 54.4% | 61.4% | 61.0% | 16.9% | 23.8% |

Table 14: Classification accuracy in Figure 3(a) with different settings. Numbers are in percentage.

|     | 5e-3          | 5e-2          | 5e-1          | 5e0           |
| --- | ------------- | ------------- | ------------- | ------------- |
| **0**   | $57.50 \pm 1.50$ | $57.51 \pm 1.63$ | $57.50 \pm 1.58$ | $55.67 \pm 2.00$ |
| **1e-3** | $57.50 \pm 1.51$ | $57.51 \pm 1.63$ | $57.50 \pm 1.58$ | $55.67 \pm 2.00$ |
| **5e-3** | $57.49 \pm 1.52$ | $57.50 \pm 1.64$ | $57.50 \pm 1.58$ | $55.67 \pm 2.00$ |
| **1e-2** | $57.55 \pm 1.50$ | $57.51 \pm 1.65$ | $57.50 \pm 1.58$ | $55.67 \pm 2.00$ |
| **5e-2** | N/A            | $57.57 \pm 1.59$ | $57.50 \pm 1.58$ | $55.67 \pm 2.00$ |
| **1e-1** | N/A            | $57.53 \pm 1.57$ | $57.50 \pm 1.59$ | $55.67 \pm 2.00$ |
| **5e-1** | N/A            | N/A            | $57.49 \pm 1.60$ | $55.67 \pm 2.00$ |
| **1e0**  | N/A            | N/A            | $57.40 \pm 1.58$ | $55.67 \pm 2.00$ |
| **5e0**  | N/A            | N/A            | N/A            | $55.76 \pm 1.86$ |

Table 15: Classification accuracy in Figure 3(b) with different settings. Numbers are in percentage.

|       | 0.6           | 0.7           | 0.8           | 0.9           |
| ----- | ------------- | ------------- | ------------- | ------------- |
| **0**    | $63.71 \pm 0.64$ | $64.03 \pm 0.66$ | $65.87 \pm 0.49$ | $64.88 \pm 0.46$ |
| $2^0$ | $63.71 \pm 0.64$ | $64.03 \pm 0.66$ | $65.87 \pm 0.49$ | $64.88 \pm 0.46$ |
| $2^1$ | $63.67 \pm 0.64$ | $64.04 \pm 0.67$ | N/A            | N/A            |
| $2^2$ | $63.69 \pm 0.67$ | N/A            | N/A            | N/A            |
| $2^3$ | N/A            | N/A            | N/A            | N/A            |
| $2^4$ | N/A            | N/A            | N/A            | N/A            |

Based on the discussion above, we would like to note that no existing work can be directly adopted to tackle the studied problem in this paper.

### C.6 ADDITIONAL EXPERIMENTS ON DIFFERENT DATASETS

To further validate the performance of the proposed method, we also perform experiments with the same commonly used popular GNN backbone models (as Section 4.2) on two Pokec datasets, namely Pokec-z and Pokec-n. We present the experimental results in Table 16, where all numerical numbers are in percentage. We observe that (1) the GNNs equipped with ELEGANT achieve comparable node classification accuracy; (2) the GNNs equipped with ELEGANT achieve consistently lower levels of bias; and (3) the values of the Fairness Certification Rate (FCR) for all GNNs equipped with ELEGANT exceed 90%, exhibiting satisfying usability. All three observations are consistent with the experimental results and the corresponding discussion presented in Section 4.2. Therefore, we argue that the effectiveness of the proposed approach is not determined by the dataset and is well generalizable over different graph datasets.

### C.7 SCALABILITY OF ELEGANT

In this subsection, we discuss the scalability of ELEGANT. Specifically, we note that if the Gaussian and Bernoulli noise is directly added over the whole graph, scaling to larger graphs would be difficult. However, the proposed approach can be easily extended to the batch training case, which has been widely adopted by existing scalable GNNs. Specifically, a commonly adopted batch training strategy of scalable GNNs is to only input a node and its surrounding subgraph into the GNN, since the prediction of GNNs only depends on the information of the node itself and its multi-hop neighbors, and the number of hops is determined by the layer number of GNNs. Since the approach proposed in our paper aligns with the basic pipeline of GNNs, the perturbation can also be performed for each specific batch of nodes. In this case, all theoretical analyses in this paper still hold, since they also do

Table 16: Experimental results on Pokec-z and Pokec-n datasets.

| | Pokec-z | | | Pokec-n | | |
|---|---|---|---|---|---|---|
| | ACC ($\uparrow$) | Bias ($\downarrow$) | FCR ($\uparrow$) | ACC ($\uparrow$) | Bias ($\downarrow$) | FCR ($\uparrow$) |
| SAGE | $63.13 \pm 0.37$ | $6.29 \pm 0.20$ | - | $57.60 \pm 2.74$ | $6.43 \pm 1.08$ | - |
| E-SAGE | $62.09 \pm 2.22$ | $4.18 \pm 1.87$ | $94.00 \pm 5.66$ | $60.74 \pm 1.87$ | $5.23 \pm 0.13$ | $91.50 \pm 7.78$ |
| GCN | $64.89 \pm 0.93$ | $3.44 \pm 0.16$ | - | $59.86 \pm 0.09$ | $4.26 \pm 0.40$ | - |
| E-GCN | $62.38 \pm 0.26$ | $1.52 \pm 0.49$ | $90.50 \pm 0.71$ | $59.83 \pm 4.16$ | $3.23 \pm 1.20$ | $94.00 \pm 8.49$ |
| JK | $63.06 \pm 1.00$ | $7.89 \pm 3.05$ | - | $57.70 \pm 1.05$ | $8.81 \pm 2.46$ | - |
| E-JK | $61.49 \pm 2.55$ | $3.63 \pm 2.18$ | $87.50 \pm 2.12$ | $61.19 \pm 0.50$ | $5.60 \pm 0.01$ | $93.00 \pm 9.90$ |

not rely on the assumption of non-batch training. Therefore, we would like to argue that the proposed approach can be easily scaled to large graphs.