# OpenReview forum: "Certified Defense on the Fairness of Graph Neural Networks"
_ICLR.cc/2025/Conference — Submitted to ICLR 2025_

### Official Review · Reviewer_X8pW · 2024-11-01

**Soundness:** 3
**Presentation:** 3
**Contribution:** 3
**Rating:** 8
**Confidence:** 4

**Summary:**

This paper proposes a framework called ELEGANT, which is designed to guarantee GNNs to be fair, even if they’re under attack within certain limits of budgets. What’s appealing about this framework ELEGANT is that it works with any GNN model as a plug-and-play method, meaning it doesn’t require re-training or making assumptions about the structure of the GNNs. This kind of flexibility and ease of use might make it a valuable tool, and it has the potential to inspire others in the field to build on its ideas.

**Strengths:**

(1) The theoretical analysis is laid out in a detailed and step-by-step manner, walking through the certification on node attributes and graphs with rigor. This could be helpful for readers trying to follow along.

(2) Unlike most existing certification methods, which don’t address fairness in GNNs, this framework is specifically aimed at fairness issues. This should be considered a significant step for the field, where fairness is a major concern but often gets sidelined.

(3) Since ELEGANT is a re-training-free, drop-in solution for GNNs, it could be easy for researchers to try it out and see the impact. Its flexibility and practicality make it stand out, and it might encourage more work along these lines.

(4) The authors evaluated the proposed framework on a solid selection of datasets and GNN models. Their experiments are thorough and show strong evidence of the effectiveness of this proposed approach.

**Weaknesses:**

(1) The general theoretical results in A.9 could use a clearer introduction earlier in the paper, to make it easier for interested readers to gain details about the theorems.

(2) There’s a mention of constraints on node attributes and graph structure that are supposed to be “unnoticeable,” but it’s not clear why these are necessary.

(3) Are there any assumptions about the types of graphs that the framework applies to? For instance, does it matter if they’re directed or undirected, or weighted or unweighted?

**Questions:**

See weaknesses.

---

> ### Author Response · Authors · 2024-11-18
> **Author Responses**
>
> We sincerely appreciate the time and efforts you've dedicated to reviewing and providing invaluable feedback. We provide a point-to-point reply below to clarify the mentioned concerns and questions.
>
> ---
>
> >  **Reviewer**: The general theoretical results in A.9 could use a clearer introduction earlier in the paper, to make it easier for interested readers to gain details about the theorems.
>
> **Authors:** We thank you for the suggestion. We will revise the discussion of our theoretical results to bring the audience's attention to Appendix A.9.
>
> ---
>
> >  **Reviewer**: There’s a mention of constraints on node attributes and graph structure that are supposed to be “unnoticeable,” but it’s not clear why these are necessary.
>
> **Authors:** We thank you for the feedback. First, we note that the constraint of unnoticeability serves as **a common setting for potential attacks** [1, 2, 3]. This is because if the attacker injects noticeable changes to the data (i.e., performs noticeable attacks), the owner of the data and the user of the model will be aware of the attacks in time, without letting the corrupted output influence the following decision-making process. This directly fails the attack. Correspondingly, in this paper, we adopt such a common setting (i.e., adding unnoticeable constraints) to formulate the potential attacks.
>
> Second, in terms of defense, the constraint of unnoticeability **strengthens the significance of obtaining perturbation-invariant budgets**. This is because if the value of the perturbation-invariant budget is greater than the constraint of unnoticeability, the GNN model is then theoretically certain to be safe in practice. This is because in such a case, the attacks will either be aware of by the user/data owner or be successfully defended. We will improve the explanation about the constraint of unnoticeability to provide a better understanding in our next version.
>
> [1] Dai, E., et al. Unnoticeable backdoor attacks on graph neural networks. TheWebConf 2023.
>
> [2] Zügner, D., et al. Adversarial attacks on graph neural networks: Perturbations and their patterns. TKDD 2020.
>
> [3] Zhang, X., et al. Gnnguard: Defending graph neural networks against adversarial attacks. NeurIPS 2020.
>
> ---
>
> >  **Reviewer**: Are there any assumptions about the types of graphs that the framework applies to? For instance, does it matter if they’re directed or undirected, or weighted or unweighted?
>
> **Authors:** We thank you for the question. We note that **ELEGANT can be applied to graphs with binary and directed edges** by removing the constraint of the added noise matrix (given by the function $\gamma_{\boldsymbol{A}}$) to be symmetric. In this case, all theoretical analysis will still hold, since all proofs do not rely on the symmetricity of such a noise matrix. For graphs with weighted edges, ELEGANT can be generalized by replacing the binary noise with a continuous noise (e.g., Gaussian noise). The strategy to calculate the perturbation-invariant budgets will be determined by the specific problem settings.
>
> ---
>
> We thank you again for your valuable feedback on our work. With the clarification above, we believe that we have responded to and addressed all your concerns with our point-to-point responses — in light of this, **we hope you consider raising your score**. Please let us know in case there are any other concerns, and if so, we would be happy to respond.

---

> > ### Comment · Reviewer_X8pW · 2024-11-22
> >
> > Thanks for your detailed response. It resolved most of my concerns. I think this paper's contribution is more than "marginally above the acceptance threshold". Since there is no 7 in ICLR, I adjusted my score to 8. I hope the discussions can be included in the next version of the paper.

---

> > > ### Author Response · Authors · 2024-11-22
> > >
> > > We are glad to hear that your concerns have been properly addressed, and we are grateful for your expertise and your dedicated efforts in helping improve our paper!

---

### Official Review · Reviewer_XZ95 · 2024-11-02

**Soundness:** 3
**Presentation:** 3
**Contribution:** 1
**Rating:** 5
**Confidence:** 4

**Summary:**

This paper proposes a fairness certification framework for Graph Neural Networks (GNNs) based on randomized smoothing. The corresponding framework, ELEGANT, provides a certain level of fairness guarantee for node classification under certain perturbation budgets.

**Strengths:**

- Paper is well-written and fluent.

- The considered research problem is a novel and important one.

- The paper provides theoretical guarantees for fairness certification.

**Weaknesses:**

- The analysis in this paper is developed on existing randomized smoothing works, where certified evaluation has already been studied for both discrete and continuous data. This paper combines such works and reframes the problem from fairness aspect, which I believe limits its novelty in terms of analysis strategy.

- As also explained in the paper, fairness improvements observed in the Experimental Results and Conclusions are probably just a by-product of the applied noise, thus there is not of sufficient contribution from the bias mitigation aspect.

- The proposed scheme cannot be directly used with fairness metrics that require ground-truth labels. However, the Authors claim that "Here bias is measured with $\Delta_{SP}$ and we have similar observations on $\Delta_{EO}$" without explaining how they used ELEGANT for the certification of $\Delta_{EO}$ (this metric requires ground-truth label information). Also, throughout the paper they never mention this limitation.

**Questions:**

Please see weaknesses.

---

> ### Author Response · Authors · 2024-11-18
> **Author Responses 1/2**
>
> We sincerely appreciate the time and efforts you've dedicated to reviewing and providing invaluable feedback. We provide a point-to-point reply below to clarify the mentioned concerns and questions.
>
> ---
>
> >  **Reviewer**: The analysis in this paper is developed on existing randomized smoothing works, where certified evaluation has already been studied for both discrete and continuous data. This paper combines such works and reframes the problem from fairness aspect, which I believe limits its novelty in terms of analysis strategy.
>
> **Authors**: We thank the reviewer for the feedback. We note that our work is fundamentally different from existing works that explore certification on accuracy, and **existing techniques cannot be directly adopted to handle the studied problem**. We provide a detailed discussion about the differences between ELEGANT and existing works on certified defense below.
>
> *Difference 1*: Most existing works (e.g., [1, 2, 3]) on certified defense for robustness are only able to achieve robustness certification over each single data point. Nevertheless, in our studied problem, certification over fairness naturally requires achieving certification over a group of nodes at the same time. This is because the level of fairness cannot be determined by the output corresponding to any single node. Such a difference necessitates the indicator function and the formulation of smoothing noise over a set of nodes in our proposed strategy.
>
> *Difference 2*: Existing works on certified defense for robustness mainly focus on i.i.d. data (e.g., images [1, 4]), while only limited works can be applied to GNNs to properly achieve certification over graph data. The links between nodes and the information propagation mechanism in GNNs make it possible for attackers to influence a large group of nodes by only attacking a few nodes in the graph. Such a difference necessitates formulating the proposed strategy based on a set of vulnerable nodes instead of only focusing on the attack over every single node.
>
> *Difference 3*: In graphs, most existing works for robustness certification (e.g., [5, 6, 7]) can only defend against attacks on either node attributes or graph structure. Due to the multi-modal input data of GNNs, existing works usually fail to handle the attacks over node attributes and graph structure at the same time. However, ELEGANT is able to defend against attacks over both data modalities. This necessitates using both continuous and discrete noises for smoothing and the certification analysis in the span of the two modalities (as shown in Figure 1).
>
> To summarize, compared with most existing works on certified defense for robustness, ELEGANT differs from other certification works on accuracy in *certification goal (Difference 1)*, *the type of data to perform certification on (Difference 2)*, and *the number of data modalities supported (Difference 3)*. We will add a more detailed discussion about the differences in our next version as suggested. We have further clarified the key novelty and contribution of our study compared with other existing works in Appendix C.5, and we hope our supplementary discussion helps address your concerns.
>
> [1] Cohen, J., Rosenfeld, E., & Kolter, Z. (2019, May). Certified adversarial robustness via randomized smoothing. In international conference on machine learning (pp. 1310-1320). PMLR.
>
> [2] Xiao, C., Chen, Z., Jin, K., Wang, J., Nie, W., Liu, M., ... & Song, D. (2022, September). Densepure: Understanding diffusion models for adversarial robustness. In ICLR.
>
> [3] Li, B., Chen, C., Wang, W., & Carin, L. (2019). Certified adversarial robustness with additive noise. NeurIPS.
>
> [4] Fischer, M., Baader, M., & Vechev, M. (2020). Certified defense to image transformations via randomized smoothing. NeurIPS.
>
> [5] Wang, B., Jia, J., Cao, X., & Gong, N. Z. (2021, August). Certified robustness of graph neural networks against adversarial structural perturbation. In SIGKDD.
>
> [6] Scholten, Y., Schuchardt, J., Geisler, S., Bojchevski, A., & Günnemann, S. (2022). Randomized message-interception smoothing: Gray-box certificates for graph neural networks. NeurIPS.
>
> [7] Geisler, S., Zügner, D., & Günnemann, S. (2020). Reliable graph neural networks via robust aggregation. NeurIPS.

---

> ### Author Response · Authors · 2024-11-18
> **Author Responses 2/2**
>
> ---
>
> >  **Reviewer**: As also explained in the paper, fairness improvements observed in the Experimental Results and Conclusions are probably just a by-product of the applied noise, thus there is not of sufficient contribution from the bias mitigation aspect.
>
> **Authors**: We thank the reviewer for pointing this out, while we feel it is necessary to point out a **misunderstanding** here: **the main focus of this study is to achieve certification over the fairness level of the GNN prediction results**.  Specifically, we emphasize that the main contribution of this paper is establishing an approach for joint certification over the span of both continuous and discrete input graph data. To the best of our knowledge, this serves as the first work that (1) achieves certified defense for the fairness of GNNs and (2) achieves joint certification against attacks on both node attributes and graph topology. Hence we argue that bringing additional benefits in bias mitigation does not jeopardize the contribution of our paper. We hope our clarification above helps address your concern.
>
> ---
>
> >  **Reviewer**: The proposed scheme cannot be directly used with fairness metrics that require ground-truth labels. However, the Authors claim that "Here bias is measured with $\Delta_{SP}$ and we have similar observations on $\Delta_{EO}$" without explaining how they used ELEGANT for the certification of $\Delta_{EO}$ (this metric requires ground-truth label information). Also, throughout the paper they never mention this limitation.
>
> **Authors**: We thank the reviewer for the feedback. We agree with the reviewer that our introduction here is not clear enough and may lead to confusion. We note that experiments based on $\Delta_{EO}$ are conducted **according to the predicted labels of a surrogate GNN model** that is separately trained on the same dataset with the same training set. We thank you again for pointing this out, and we will elaborate on the experimental setting differences under the two metrics in Section 4 to avoid confusion.
>
> ---
>
> We thank you again for your valuable feedback on our work. With the clarification above, we believe that we have responded to and addressed all your concerns with our point-to-point responses — in light of this, **we hope you consider raising your score**. Please let us know in case there are any other concerns, and if so, we would be happy to respond.

---

> ### Author Response · Authors · 2024-11-25
> **A Kind Reminder**
>
> Dear Reviewer XZ95,
>
> Thank you for your valuable feedback on our work. We have prepared a thorough response to address your concerns. We believe that we have responded to and addressed all your concerns with our responses — in light of this, **we hope you consider raising your score**.
>
> Notably, given that we are approaching the deadline for the rebuttal phase, we hope we can receive your feedback soon. We thank you again for your efforts and suggestions!

---

> ### Author Response · Authors · 2024-11-27
>
> We are glad to share that we have achieved a consensus with all other three reviewers on their positive ratings, and we also appreciate your highly positive feedback on our soundness and presentation!
>
> However, considering that you have not replied to our carefully formulated point-to-point responses, we kindly reach out to request your feedback. We believe that we have responded to and addressed all your concerns, and hence we hope you consider raising your score.
>
> We thank you again for your efforts and suggestions, and we look forward to your feedback!

---

> ### Author Response · Authors · 2024-12-01
> **A Kind Reminder for the Last Two Days**
>
> We sincerely thank you for your efforts in helping us improve our paper! With only two days remaining and no response yet to our carefully formulated point-to-point replies, we are kindly reaching out to request your feedback. We believe we have thoroughly addressed all your concerns and respectfully hope you **consider raising your score**.
>
> Once again, we deeply appreciate your efforts and valuable suggestions, and we look forward to your feedback!

---

### Official Review · Reviewer_zaDM · 2024-11-03

**Soundness:** 3
**Presentation:** 3
**Contribution:** 2
**Rating:** 6
**Confidence:** 4

**Summary:**

This paper introduces ELEGANT, a framework for certifiable defense on the fairness of GNNs against adversarial perturbations in node attributes and graph topology. ELEGANT aims to serve as a plug-and-play module that can be applied to any optimized GNN model without altering the underlying structure or requiring retraining. The paper's contributions include a formal problem formulation, algorithmic design leveraging randomized smoothing, and extensive experiments to validate the approach across real-world datasets.

**Strengths:**

- The authors provide comprehensive experimental validation across multiple real-world datasets, demonstrating the generalizability and effectiveness of ELEGANT across different GNN architectures.
- The paper is well-written, with clear explanations of the methodology and experimental setup.

**Weaknesses:**

- The setting, though well-defined theoretically, may lack practical applicability in real-world scenarios. The assumptions about the perturbation budgets and fairness certification requirements may not always align with practical needs, which could limit the framework’s deployment in real-life applications. The practical feasibility of maintaining certified fairness under realistic conditions could be further substantiated.
- The Monte Carlo estimation and probabilistic guarantee calculation introduce added complexity to the certification process. While theoretically sound, this approach could pose challenges in implementation, especially in large-scale networks. Further simplification or performance analysis would strengthen the practical value of the proposed solution.

**Questions:**

Please refer to the weakness.

---

> ### Author Response · Authors · 2024-11-18
> **Author Responses 1/2**
>
> We sincerely appreciate the time and efforts you've dedicated to reviewing and providing invaluable feedback. We provide a point-to-point reply below to clarify the mentioned concerns and questions.
>
> ---
>
> >  **Reviewer**: The setting, though well-defined theoretically, may lack practical applicability in real-world scenarios. The assumptions about the perturbation budgets and fairness certification requirements may not always align with practical needs, which could limit the framework’s deployment in real-life applications. The practical feasibility of maintaining certified fairness under realistic conditions could be further substantiated.
>
> **Authors**: We thank the reviewer for the feedback. We would like to highlight two key perspectives of real-world application scenarios below to further substantiate the realistic use of our study.
>
> First, there are **unintentional cases where perturbations (e.g., noise) may be made to the data and finally lead to corrupted fairness levels**, e.g., human annotators could systematically make mistakes when assigning labels to images of people of a certain skin color [1]. Such situations can also happen in graph machine learning scenarios and finally collapse the fairness in high-stake decision-making scenarios, e.g., fraud detection in financial transaction networks can favor individuals from certain communities. We note that the strength of noise of these scenarios often falls into a certain range, which can be considered as the budget in our work to achieve protection on the fairness of the GNN models. In such cases, our work can be directly adopted to achieve such certified protection without the need for re-training.
>
> Second, there has been plenty of works focusing on the fairness attack and defense problems of machine learning algorithms over traditional i.i.d. data, e.g., [1, 2, 3, 4, 5]. **Such a problem has been widely acknowledged to be critical in real-world applications**. For example, the developer of a criminal recidivism prediction model could manipulate the data in a discriminatory manner to bias the decision against a certain group of people [1, 6]; in credit or loan applications, adversaries can even profit from such attacks by biasing decisions for their own benefit or depreciating the value and credibility of government and agencies [2]. Especially, graph data has become a key format of data to characterize the complex relationship between data points in these applications, and thus achieving certification for graph machine learning bears critical practical significance under these realistic conditions. Our work can also be directly adopted in these applications to protect fairness, which helps enhance the authority of decision-making agencies and protect the benefit of underrepresented communities.
>
> We thank you again for bringing this up, and we will revise our paper to emphasize the practical significance of our study by adding the discussion presented above. We hope our supplementary discussion addressed your concerns.
>
> [1] Solans, D., Biggio, B., & Castillo, C. (2020). Poisoning attacks on algorithmic fairness. In ECML-PKDD.
>
> [2] Mehrabi, N., Naveed, M., Morstatter, F., & Galstyan, A. (2021). Exacerbating algorithmic bias through fairness attacks. In AAAI.
>
> [3] Van, M. H., Du, W., Wu, X., & Lu, A. (2022). Poisoning attacks on fair machine learning. In DASFAA.
>
> [4] Chhabra, A., Singla, A., & Mohapatra, P. (2021). Fairness degrading adversarial attacks against clustering algorithms. arXiv preprint arXiv:2110.12020.
>
> [5] Jo, C., Sohn, J. Y., & Lee, K. (2022). Breaking fair binary classification with optimal flipping attacks. In ISIT.
>
> [6] Angwin, J., Larson, J., Mattu, S., & Kirchner, L. (2022). Machine bias. In Ethics of data and analytics (pp. 254-264). Auerbach Publications.

---

> ### Author Response · Authors · 2024-11-18
> **Author Responses 2/2**
>
> ---
>
> >  **Reviewer**: The Monte Carlo estimation and probabilistic guarantee calculation introduce added complexity to the certification process. While theoretically sound, this approach could pose challenges in implementation, especially in large-scale networks. Further simplification or performance analysis would strengthen the practical value of the proposed solution.
>
> **Authors**: We thank the reviewer for the feedback. We agree with the reviewer that Monte Carlo estimation can bring concerns about time complexity. Therefore, we have integrated a **carefully designed method to reduce the time complexity** as introduced in Appendix B.6, which significantly reduces the actual running time in our experiments. Meanwhile, since the proposed approach **does not involve any re-training process** (which is actually the most costly part), the actual running time is satisfying according to the empirical results shown in Appendix B.8. Finally, we have also elaborated on the discussion of the scalability of the proposed method in Appendix C.7.
>
> We thank you again for bringing this up, and we will elaborate on the discussion about time complexity reduction and running time performance in our paper. We hope our supplementary discussion addressed your concerns.
>
> ---
>
> We thank you again for your valuable feedback on our work. With the clarification above, we believe that we have responded to and addressed all your concerns with our point-to-point responses — in light of this, **we hope you consider raising your score**. Please let us know in case there are any other concerns, and if so, we would be happy to respond.

---

> ### Author Response · Authors · 2024-11-25
> **A Kind Reminder**
>
> Dear Reviewer zaDM,
>
> Thank you for your valuable feedback on our work. We have prepared a thorough response to address your concerns. We believe that we have responded to and addressed all your concerns with our responses — in light of this, **we hope you consider raising your score**.
>
> Notably, given that we are approaching the deadline for the rebuttal phase, we hope we can receive your feedback soon. We thank you again for your efforts and suggestions!

---

> > ### Comment · Reviewer_zaDM · 2024-11-25
> > **Official Comment by Reviewer zaDM**
> >
> > Some of my concerns have been addressed, and I appreciate the efforts made to clarify and improve the paper. As a result, I have raised my score. Thank you for your responsiveness.

---

> > > ### Author Response · Authors · 2024-12-01
> > >
> > > We are glad to hear that your concerns have been properly addressed, and we are grateful for your expertise and your dedicated efforts in helping improve our paper!

---

### Official Review · Reviewer_zjmd · 2024-11-04

**Soundness:** 3
**Presentation:** 3
**Contribution:** 2
**Rating:** 8
**Confidence:** 4

**Summary:**

This works aims to formulate the problem of certified defense on the fairness. To achieve this goal, the certified robustness analysis is applied to the certified fairness against input perturbations. More specifically, the analysis about certified fairness defense is conducted on node attributes and graph structures. Experiments are conducted on three real-world datasets.

**Strengths:**

1. This work focuses on an important problem of fairness on graph-structured data. Many real-world applications such as financial analysis and social network analysis requires the fairness to be guaranteed.
2. The authors propose a novel problem of certified fairness defense to avoid the attacks on the fairness.
3. The analysis in certified fairness defense is solid. And surprisingly, the certified fairness method empirically improve the fairness.

**Weaknesses:**

Overall, the reviewer feel the strengths outweigh the weaknesses. However, several concerns would be highly suggested to be addressed.
1. The certified fairness defense is an application of certified robustness by changing the accuracy metric to the fairness metric in the analysis. This is quite straightforward. Hence, the contributions in theoretical analysis is somewhat limited.
2. For certified robustness methods, they improve the robustness because of the adversarial training stage of the trained model. For this proposed method, bias is decreased without explicit way to improve the fairness. It is highly suggested to conduct some further theoretical analysis for this interesting phenomenon.
3. Experiments on large graphs with none-synthetic edges are suggested.

**Questions:**

Please refer to the weakness.

In addition. for certified adversarial robustness methods, they generally are only applicable to the evasion attack, where the perturbations are only conducted on the test samples. For the proposed method, can it defend the poisoning attack?

---

> ### Author Response · Authors · 2024-11-18
> **Author Responses 1/2**
>
> We sincerely appreciate the time and efforts you've dedicated to reviewing and providing invaluable feedback. We provide a point-to-point reply below to clarify the mentioned concerns and questions.
>
> ---
>
> >  **Reviewer**: The certified fairness defense is an application of certified robustness by changing the accuracy metric to the fairness metric in the analysis. This is quite straightforward. Hence, the contributions in theoretical analysis is somewhat limited.
>
> **Authors**: We thank the reviewer for the feedback. We note that our work is fundamentally different from existing works that explore certification on accuracy, and **existing techniques cannot be directly adopted to handle the studied problem**. We provide a detailed discussion about the differences between ELEGANT and existing works on certified defense below.
>
> *Difference 1*: Most existing works (e.g., [1, 2, 3]) on certified defense for robustness are only able to achieve robustness certification over each single data point. Nevertheless, in our studied problem, certification over fairness naturally requires achieving certification over a group of nodes at the same time. This is because the level of fairness cannot be determined by the output corresponding to any single node. Such a difference necessitates the indicator function and the formulation of smoothing noise over a set of nodes in our proposed strategy.
>
> *Difference 2*: Existing works on certified defense for robustness mainly focus on i.i.d. data (e.g., images [1, 4]), while only limited works can be applied to GNNs to properly achieve certification over graph data. The links between nodes and the information propagation mechanism in GNNs make it possible for attackers to influence a large group of nodes by only attacking a few nodes in the graph. Such a difference necessitates formulating the proposed strategy based on a set of vulnerable nodes instead of only focusing on the attack over every single node.
>
> *Difference 3*: In graphs, most existing works for robustness certification (e.g., [5, 6, 7]) can only defend against attacks on either node attributes or graph structure. Due to the multi-modal input data of GNNs, existing works usually fail to handle the attacks over node attributes and graph structure at the same time. However, ELEGANT is able to defend against attacks over both data modalities. This necessitates using both continuous and discrete noises for smoothing and the certification analysis in the span of the two modalities (as shown in Figure 1).
>
> To summarize, compared with most existing works on certified defense for robustness, our study differs from other certification works on accuracy in *certification goal (Difference 1)*, *the type of data to perform certification on (Difference 2)*, and *the number of data modalities supported (Difference 3)*. We will add a more detailed discussion about the differences in our next version as suggested. We have further clarified the key novelty and contribution of our study compared with other existing works in Appendix C.5, and we hope our supplementary discussion helps address your concerns.
>
> [1] Cohen, J., Rosenfeld, E., & Kolter, Z. (2019, May). Certified adversarial robustness via randomized smoothing. In international conference on machine learning (pp. 1310-1320). PMLR.
>
> [2] Xiao, C., Chen, Z., Jin, K., Wang, J., Nie, W., Liu, M., ... & Song, D. (2022, September). Densepure: Understanding diffusion models for adversarial robustness. In ICLR.
>
> [3] Li, B., Chen, C., Wang, W., & Carin, L. (2019). Certified adversarial robustness with additive noise. NeurIPS.
>
> [4] Fischer, M., Baader, M., & Vechev, M. (2020). Certified defense to image transformations via randomized smoothing. NeurIPS.
>
> [5] Wang, B., Jia, J., Cao, X., & Gong, N. Z. (2021, August). Certified robustness of graph neural networks against adversarial structural perturbation. In SIGKDD.
>
> [6] Scholten, Y., Schuchardt, J., Geisler, S., Bojchevski, A., & Günnemann, S. (2022). Randomized message-interception smoothing: Gray-box certificates for graph neural networks. NeurIPS.
>
> [7] Geisler, S., Zügner, D., & Günnemann, S. (2020). Reliable graph neural networks via robust aggregation. NeurIPS.

---

> ### Author Response · Authors · 2024-11-18
> **Author Responses 2/2**
>
> ---
>
> >  **Reviewer**: For certified robustness methods, they improve the robustness because of the adversarial training stage of the trained model. For this proposed method, bias is decreased without explicit way to improve the fairness. It is highly suggested to conduct some further theoretical analysis for this interesting phenomenon.
>
> **Authors**: We thank the reviewer for pointing this out, and we agree with the reviewer that this is an important phenomenon our audience can be interested in. We note that achieving better fairness levels is a byproduct of our proposed method. To clarify this, **we showed analysis in Appendix B.9**. A key reason is that the distribution difference in the node attribute values and edge existence across different subgroups is a significant source of bias [1]. Adding noise on both node attributes and graph topology can reduce distributional divergence and thus mitigate bias. As suggested, we will conduct further theoretical analysis to investigate this phenomenon and add the corresponding discussion to our work. We thank you again for the feedback, and we hope the discussion above addressed your concerns.
>
> [1] Fan, W., et al. Fair graph auto-encoder for unbiased graph representations with Wasserstein distance. ICDM 2021.
>
> ---
>
> >  **Reviewer**: Experiments on large graphs with non-synthetic edges are suggested.
>
> **Authors**: We thank the reviewer for the feedback. We agree with the reviewer that conducting experiments based on larger graphs with non-synthetic edges will further validate the effectiveness of the proposed method and enhance our conclusions. Therefore, **we have presented comprehensive experimental results on two larger datasets with real-world connections**, including Pokec-z and Pokes-n, under multiple popular GNN backbones, and we have presented these **supplementary results in Appendix C**, including the discussion on the scalability of the proposed method in Appendix C.7. According to the experimental results, we found that the conclusions remain consistent with those introduced in our Section 4, and we hope these additional experiments addressed your concerns.
>
> ---
>
> >  **Reviewer**: In addition. for certified adversarial robustness methods, they generally are only applicable to the evasion attack, where the perturbations are only conducted on the test samples. For the proposed method, can it defend the poisoning attack?
>
> **Authors**: We thank the reviewer for the feedback. We note that the main goal of this work is to certifiably defend any optimized GNNs against perturbations that may compromise fairness (as defined in Problem 1), ensuring broad applicability such that the proposed approach can be directly adopted by researchers and practitioners without re-training. This consideration aligns with a widely acknowledged consensus in this line of research. Thus, addressing poisoning attacks is not the focus of our proposed defense method, nor is it the focus of most other existing works on certified defense.
>
> Although the setting of poisoning attacks is not involved in the formulation of Problem 1, we agree with the reviewer that achieving certified defense against poisoning attacks is interesting and worth exploring in our future works.
>
> ---
>
> We thank you again for your valuable feedback on our work. With the clarification above, we believe that we have responded to and addressed all your concerns with our point-to-point responses — in light of this, **we hope you consider raising your score**. Please let us know in case there are any other concerns, and if so, we would be happy to respond.

---

> ### Author Response · Authors · 2024-11-25
> **A Kind Reminder**
>
> Dear Reviewer zjmd,
>
> Thank you for your valuable feedback on our work. We have prepared a thorough response to address your concerns. We believe that we have responded to and addressed all your concerns with our responses — in light of this, **we hope you consider raising your score**.
>
> Notably, given that we are approaching the deadline for the rebuttal phase, we hope we can receive your feedback soon. We thank you again for your efforts and suggestions!

---

> > ### Comment · Reviewer_zjmd · 2024-12-01
> > **I have increased the score**
> >
> > Dear Authors,
> >
> > Sorry for the late response. Your rebuttal address most of my questions. There is just one question I really hope it could be addressed in the future, i.e., the theoretical analysis about why certified fairness can improve the fairness. I agree with another reviewer that 7 would a fair score. But considering there is no 7, I have increased the score to 8.
> >
> > Bests,
> > Reviewer

---

> > > ### Author Response · Authors · 2024-12-01
> > >
> > > We are glad to hear that most of your concerns have been properly addressed! We would like to kindly remind you that the current rating remains at 6, and we sincerely hope you consider revising it at your earliest convenience.
> > >
> > > We are truly grateful for your expertise and your dedicated efforts in helping improve our paper, and we will extend Appendix B.9 (mentioned in our second response) with more theoretical analysis to deliver a clearer understanding of improving fairness with our proposed ceertification method. We thank you again for your feedback.

---

> > > ### Author Response · Authors · 2024-12-01
> > >
> > > We confirm that the rating has been successfully revised. We thank you again for your your expertise and your dedicated efforts in helping improve our paper!

---

### Comment · Area_Chair_6WVN · 2024-11-28

I would like to encourage the reviewers to engage with the author's replies if they have not already done so. At the very least, please
acknowledge that you have read the rebuttal.

---

### Meta-Review · Area_Chair_6WVN · 2024-12-21

**Metareview:**

The paper provide a certificate whether a given fairness metric (e.g. SP or EO) is below some specified threshold. It combines Gaussian smoothing for continuous data and discrete smoothing for the graph structure.

The biggest weakness of the paper is that prior work is not properly credited. Theorem 1 is a direct application of Theorem 1 in [1] (see Eq. 3) for a new classifier where the output in an. indicator function w.r.t. the bias threshold. The core result has been proven not only in [1] but also several times in follow-up papers and does not need to be re-proven! Of course, for a solid theoretical analysis one can prove that Theorem 1 from [1] is applicable, but e.g. Lemma A1 and Lemma A2 are already known. Similarly, Theorem 2 and Theorem 3 can be derived as a simple application of theorems in prior work (see [2] and the generalisation in [3]) under the new fairness setting. The fact that you can arrive at the results in this paper by applying existing theorems from prior work is not a downside, however, omitting or ignoring the contributions of prior work verges on scientific misconduct. The authors should be aware of this issue since it has been pointed out to them during the reviewing process in a prior submission of the same paper, yet they have not updated their work to reflect this.

Similarly, Reviewer zjmd also points out that "The certified fairness defense is an application of certified robustness by changing the accuracy metric to the fairness metric in the analysis. This is quite straightforward. Hence, the contributions in theoretical analysis is somewhat limited." The "differences" provided by the authors in response are not convincing (see next paragraph). The major contribution of the paper in my opinion is deriving certificates that can handle both continuous and discrete smoothing.

In the rebuttal the authors state "Most existing works (e.g., [1, 2, 3]) on certified defense for robustness are only able to achieve robustness certification over each single data point. Nevertheless, in our studied problem, certification over fairness naturally requires achieving certification over a group of nodes at the same time." Collective certificates are precisely dealing with "certification over a group of nodes". Yet, the paper on collective certificates (Schuchardt et al., 2020) is only cited in passing and the there is no discussion on the differences between the two approaches, and importantly, no comparison to the collective certificates approach which can (in theory) be adapted as a baseline to certify fairness. Similar to before, this again shows a lack of proper discussion of prior work.

As pointed out by Reviewer XZ95, another weakness is that "The proposed scheme cannot be directly used with fairness metrics that require ground-truth labels." In reply, the authors state that they use predicted labels of a surrogate GNN model, however, this is no longer certifying EO but rather some surrogate metric.

I can only recommend acceptance if the authors update the paper to fix the above issues. To be clear, I think the method itself is worth publishing but the paper in its current form is highly misleading about its contributions.

References:
1. Cohen et al. "Certified adversarial robustness via randomized smoothing"
2. Lee et al. "Tight Certificates of Adversarial Robustness for Randomly Smoothed Classifiers"
3. Bojchevski et al. "Efficient robustness certificates for discrete data: Sparsity-aware randomized smoothing for graphs, images and more"

**Additional Comments On Reviewer Discussion:**

The authors were able to adequately address most the concerns of the reviewers, with the exception of the issue w.r.t. certifying EO (see comment above).

---

### Decision · Program_Chairs · 2025-01-22

Reject